# sCellST predicts single-cell gene expression from H& E images

Loïc Chadoutaud[1,2,3], Marvin Lerousseau[1,2,3,8], Daniel Herrero-Saboya [1,2,3,4], Julian Ostermaier[1,2,3], Jacqueline Fontugne[5,6,7], Emmanuel Barillot [1,2,3] ✉ & Thomas Walter [1,2,3] ✉

Understanding the spatial organization of individual cell types within tissue and how this organization is disrupted in disease, is a central question in biology and medicine. Hematoxylin and eosin-stained slides are widely available and provide detailed morphological context, while spatial gene expression profiling offers complementary molecular insights, though it remains costly and limited in accessibility. Predicting gene expression directly from histological images is therefore an attractive goal. However, existing approaches typically rely on small image patches, limiting resolution and the ability to capture fine-grained morphological variation. Here, we introduce a deep learning approach that predicts single-cell gene expression from morphology, matching patch-based methods on spot level prediction tasks. The model recovers biologically meaningful expression patterns across two cancer datasets and distinguishes fine cell populations. This approach enables molecular-level interpretation of standard histological slides at scale, offering new opportunities to study tissue organization and cellular diversity in health and disease.

Tissue spatial organization is a fundamental feature of multicellular organisms, essential for proper development, homeostasis, and tissue repair. It depends on the precise arrangement of different cell types and their states, allowing tissues and organs to perform their biological functions efficiently. Disruptions or perturbations in this organization can lead to pathological conditions, such as cancer, where tissue structure and function become compromised.

The most widely used technique to study tissue architecture and its disease related alterations involves Hematoxylin and Eosin (H&E)-stained tissue slides which are routinely produced in clinical practice and examined by pathologists to evaluate disease states and inform treatment decisions. With the advent of machine learning for pathological slides, algorithms have been developed to address tasks such as molecular phenotyping and biomarker discovery[1]. H&E slides offer a detailed view of tissue, capturing both individual cellular phenotypes and overall architecture.

A fundamental step in tissue analysis is the identification of distinct cell types within the tissue. Broad categories, such as epithelial cells, fibroblasts or lymphocytes can be readily identified through manual inspection, due to their characteristic nuclear size, morphology and color. However, more subtle cell types often require molecular staining, such as Immunohistochemistry (IHC) or Immunofluorescence (IF), as differential transcriptional programs do not always leave visible morphological fingerprints. In some cases, cell type specific morphological cues may simply remain undiscovered.

More recently, spatial transcriptomic (ST) technologies have emerged as a powerful tool to study gene expression (GE) in the spatial context[2,3]. These technologies offer complementary analyses to H&E staining by providing insights into the molecular landscape of tissues that are neither accessible through conventional histology nor through IHC or IF technologies, which are limited in the number of molecular markers. ST technologies can be broadly divided into two

[1]Centre for Computational Biology (CBIO), Mines Paris, PSL University, Paris, France. [2]Institut Curie, Paris, France. [3]INSERM, U1331, Paris, France. [4]Computational Medicine, Servier Research & Development, Saclay, France. [5]Institut Curie, Department of Pathology, Saint-Cloud, France. [6]Institut Curie, CNRS, UMR144, Equipe labellisée Ligue Contre le Cancer, PSL Research University, Paris, France. [7]Université Paris-Saclay, UVSQ, Montigny-le-Bretonneux, France. [8]Present address: Spotlight Medical, Paris, France. ✉e-mail: emmanuel.barillot@curie.fr; thomas.walter@minesparis.psl.eu

main categories. Image-based ST such as MERFISH, COSMIX and Xenium[4] rely on fluorescence imaging and enable the capture of hundreds of different RNA species with highly precise spatial resolution, down to sub-cellular level. However, these methods do not measure the full transcriptome. On the other hand, sequencing-based methods such as Visium, Slide-seq or Stereo-seq use spatial barcodes to retain spatial information at specific locations called spots. The downside of these approaches is that spots usually contain several cells, around 10 to 20 in the case of Visium. Deconvolution algorithms[5,6] are then required to estimate cell type proportions within each spot, with very few of these approaches achieving true single cell resolution[7]. In addition, both technologies share the downside of being very expensive, which also prevent them from being used in larger cohorts, unlike H&E slides.

Recent studies have demonstrated that H&E-stained slides contain valuable information that can be leveraged to predict GE using machine learning algorithms. One pioneering method in this field is HE2RNA[8], a deep learning model designed to predict GE from bulk RNA sequencing data by aggregating predictions from small patches extracted from corresponding H&E images. Despite the inherent challenges posed by weak and noisy signals, HE2RNA effectively identifies immune cell-enriched regions, suggesting that the link between morphology and GE is sufficiently strong for this kind of approaches. With the advent of ST technologies, particularly Visium, which provides both H&E slides and transcriptomic spot data from the same tissue section, these models have gained even greater predictive power. Two types of approaches have been recently developed: super-resolution models and GE predictors. The former takes image features and spot GE as input to produce super-resolved expression maps[9]. While these models thus virtually increase the resolution inside and

between the spots, they still require ST as inputs. The latter refers to models trained to predict GE based on the image centered at each spot locations[10–17]. Other methods[18–20] predict GE at finer spatial resolutions using a weakly supervised learning approach. Weak supervision corresponds to a scenario, where the ground truth is not available for each input instance, but only for groups of instances. While these approaches enhance spatial resolution, they have not yet been convincingly shown to provide estimations of single-cell GE.

Here, we introduce sCellST, a method to predict single-cell GE from cell morphology, trained on paired ST and H&E slides. Our method is versatile and can be applied across different tissues and cancer types, as demonstrated with the diverse datasets used in this study. While designed to make cell-level rather than spot-level predictions, we show that sCellST performs on par with state-of-the-art spot-based predictors on a Kidney and a Prostate cancer datasets. Next, we validated sCellST's ability to accurately predict cell types by benchmarking against methods trained on manually annotated images and found remarkable agreement. We also compared with Xenium measurements and found strong correlations despite important technological differences. Finally, we demonstrate that sCellST can help identifying morphological patterns in more subtle cell types by leveraging a list of marker genes extracted from a scRNA-seq dataset.

## Results
### sCellST overview
We present sCellST, a weakly supervised learning framework that learns a model to predict GE from H&E images only. Our approach is composed of 3 main steps (Fig. 1).

First, a deep learning model is used to perform nucleus detection on the underlying H&E images from the spatial transcriptomic slide.

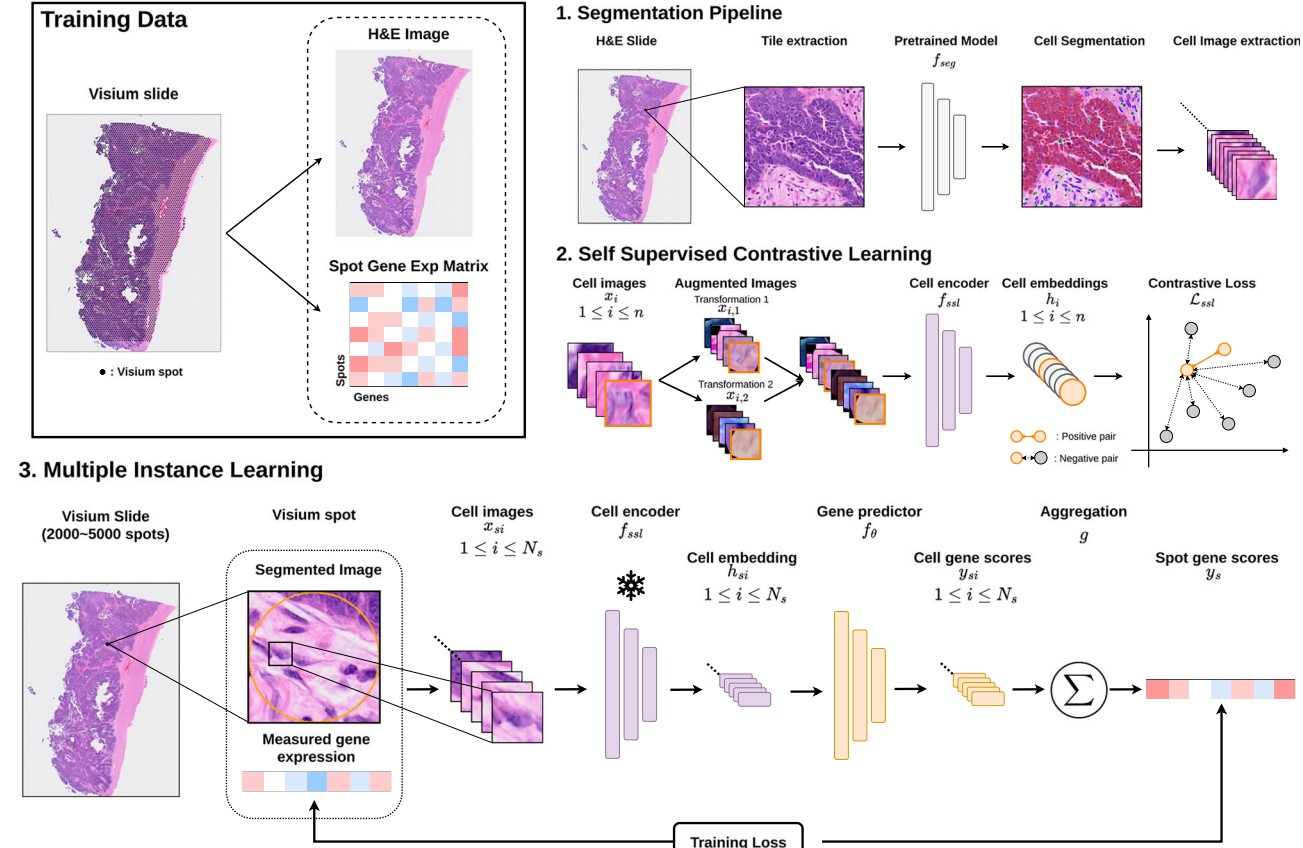

**Fig. 1 | Overview of the sCellST pipeline. 1** sCellST first uses a pretrained nuclei detection algorithm to extract cell images from H&E images. **2** A feature extractor is trained using contrastive learning on the cell images. **3** A Multiple Instance Learning approach is used to learn a gene expression (GE) predictor. The model predicts first GE score for each cell within a spot before aggregating the prediction to produce a spot expression vector which is used for training.

For each cell detected by the model, we extracted a squared crop image centered on the segmentation output masks. We used CellViT[21] for our experiments but any other detection algorithm for histopathology data ([22]) can be used.

The next step is to find a suitable representation of the cell image. For this, we turned to Self Supervised Learning (SSL), a state-of-the-art method to learn powerful embeddings which can then be used for a large variety of tasks[23]. SSL relies on the definition of pretext tasks, i.e., tasks that are not directly related to the classification or regression problem we want to solve, but for which large unlabeled datasets are available. It has shown great success on tissue patches and full slides[24] for GE predictions, tile classification or survival predictions. More recently SSL has been used to represent single cell morphologies[25,26] with great success in distinguishing different cell types. Here, we trained a MoCo v3 algorithm[27] (methods).

Finally, we trained a GE predictor using previously obtained cell embeddings. Since ground-truth GE is not available for individual cells, but rather for spots containing multiple cells, we formulated the problem as a Multiple Instance Learning (MIL) problem[28]. Specifically, for each spot in a Visium slide, we predict a GE vector for each cell within the spot. We then aggregated these predictions to generate a spot-level prediction, which is compared with the measured expression to train the algorithm (methods). The predicted cell-level scores can then be interpreted as cell GE estimates.

After training, sCellST can be applied using only H&E slides to produce single-cell and spatially resolved GE data.

## sCellST predicts single cell level gene expression in simulated data

As our datasets do not contain detailed groundtruth on single-cell GE, validation of sCellST at the cellular level is a challenging endeavor. We therefore designed several levels of validation, the first of which is based on simulation experiments using cell images from an ovarian cancer slide.

To establish a ground truth for single-cell GE, we used an annotated scRNA-seq dataset as a reference. We matched patches of detected cells with GE profiles from this scRNA-seq data according to three rules, each representing a different scenario. We note that the ground truth GE assignments used throughout simulation do not reflect real biological correspondence, but rather serve as a means to generate validation data, allowing us to assess whether the algorithmic strategy can work in principle.

In the *random* scenario, GE vectors were randomly assigned to cell images, and there was thus no relationship between morphology and GE, providing a random baseline. To artificially introduce a relationship between cell morphology and GE, we first clustered the cell image vectors. Each image cluster was then arbitrarily assigned to one scRNA-seq cluster. In the *centroid* scenario, we assigned to each cell image the mean expression vector of its corresponding scRNA-seq cluster. This scenario corresponds to an idealized case where all GE of all cells within the same cluster are equal, and there is thus no intra-cluster GE variability. Finally, we explored a more challenging scenario where we matched cells from the scRNA-seq and from the corresponding morphological cluster (*cell* scenario; see Fig. 2a). In this case, as we draw cells randomly from both matched clusters, GE and morphological intra-cluster variation are independent by construction while inter-cluster variation is perfectly aligned. In both the *centroid* and the *cell* scenario, we assume that cell type clusters are matched by distinct morphological clusters, which is an idealized hypothesis. However, in both cases, any potential links between subtle transcriptomic and morphological differences are broken by construction.

Once the cell images and GE vectors were matched, we simulated spot-level GE by summing the GE of single cells assigned to each spot (methods).

We first compared the three scenarios on the top 1000 highly variable genes (HVGs). For both spot and cell-level predictions, genes predicted in the random scenario produced a mean correlation of 0., as expected in the absence of links between GE and cellular morphology. In contrast, the *centroid* scenario yielded a mean correlation of 0.93, demonstrating the effectiveness of the MIL approach.

The *cell* scenario also resulted in a positive mean correlation (0.20). Notably, when trained only on the marker genes, which are supposed to have stronger links with morphological properties, we saw an increase in mean correlation from 0.20 for HVGs to 0.68 for marker genes (Fig. 2c) at the spot level. This indicates that the model can capture varying degrees of correlation between cell morphology and GE.

However, defining an upper bound for the evaluation metric is challenging, as even a model with access to all information would not achieve a correlation of 1 in the cell scenario, as by design of our simulation, GE is not entirely predictable from morphology. To illustrate this, we trained a model under full supervision to predict the matched GE vectors from the corresponding cell images. The performance of the supervised model was slightly higher than that of the MIL-trained model (Fig. 2d), yet still far from reaching perfect correlation (mean of 0.20 and 0.78 for HVGs and MG respectively).

Next, we investigated the effect of the tile encoding strategy by replacing the SSL-embeddings by one-hot encoded vectors of the image clusters. These one-hot-vectors represent an ideal noise-free embedding that perfectly represents the cell image clusters. We only observed a mild improvement (Fig. 2e), suggesting that the high dimension of the cell embeddings does not negatively impact prediction performance.

With these experiments, we demonstrated that a MIL approach can recover GE vectors from cell images, even in challenging settings where ground truth labels for all cell images are unavailable and the link between morphology and GE is only partial.

## sCellST performs competitively with other algorithms for spot-level predictions

Next, we aimed to benchmark sCellST against state-of-the-art methods for GE prediction from H&E data. As there is currently no method that specifically adresses single-cell GE prediction from Visium data, we performed spot-level comparisons, even though this was not the primary objective of our method. We compared our algorithm to four other methods: HisToGene[17], THItoGene[29], MCLStExp[30] and Istar[20]. HisToGene is based on a Vision Transformer neural network which takes the image from each spot as input. THItoGene builds upon the same architecture but also introduces capsule network and graph attention layer. MclSTExp is based on a contrastive learning framework similar to CLIP[31]. Since it does not natively allow to predict gene expression, the authors used a weighted KNN approach to predict GE in the inference stage. Similarly to sCellST, Istar also utilizes weakly supervised training, processing small patches from the spot images and predicting GE for each patch, which are then aggregated at the spot level.

We conducted our experiments using kidney[32] (8 Visium slides, Fig. 3b) and prostate (5 Visium slides, Fig. 3c) cancer datasets, accessed through the HEST database[33]. To benchmark the spot-level GE predictions, we used 50 and 500 HVGs and spatially variable genes (SVGs). Predictions were evaluated using Pearson and Spearman correlation coefficients. Model training was performed on either a single slide or all but one slide, with the remaining slides used for evaluation (Fig. 3a).

We first examined the top 50 HVG and SVG, where our method outperformed the others in 8 out of 8 comparisons, highlighting its effectiveness. When expanding the analysis to the top 500 HVGs and SVGs, our method remained among the top performers but showed slightly lower performance than MclStExp while still surpassing transformer-based methods. The lower performance of the latter methods might be attributed to the large number of parameters in transformer architectures relative to the limited amount of training data available.

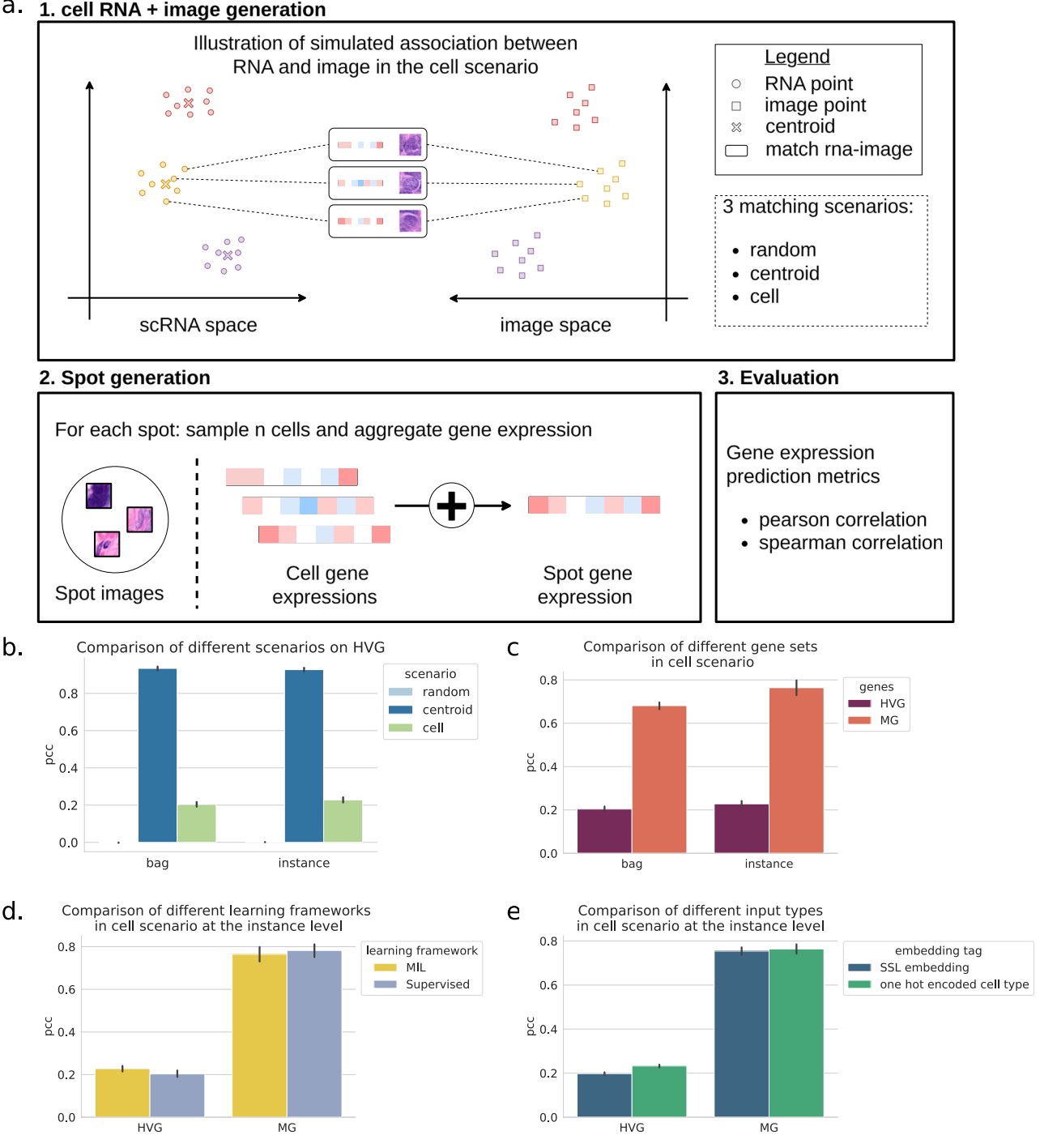

**Fig. 2 | Simulation experiments. a** Simulation framework illustrating cell image and gene expression (GE) attribution under the cell scenario; other scenarios are presented in the SupFig.10. **b**–**e** Mean Pearson correlation for different sets of genes in the test dataset (error bars correspond to 95% confidence interval obtained via bootstrapping). The number of highly variable genes (HVG) is 1000 and the number of marker genes (MG) is 51. **b** Comparison across different scenarios (cell, centroid, random) at the bag and instance level on HVG. **c** Comparison across different sets of genes (HVG or MG) at the bag and instance level in the cell scenario. **d** Comparison across different learning frameworks (MIL or supervised) on different sets of genes (HVG and MG) in the cell scenario. **e** Comparison across different instance encodings (derived from SSL embeddings or one hot encoded cell types) on different sets of genes (HVG and MG) in the cell scenario.

We provide additional experiments about the impact of gene number on the performances of our model in SupFig. 1. On the Prostate dataset in the multiple slide training case, we found that although the best predicted genes are among the top 200 HVG there is not a clear link between gene predictability and gene rank.

Furthermore, we investigated whether the choice of embeddings impacts the overall results, and we found that our SSL embeddings are better than the ones derived by pre-training on ImageNet (mean PCC 0.16 versus 0.13, SupFig.2).

Overall, our experiments demonstrate that sCellST performs competitively with the best existing methods in most settings. This result was unexpected, given that our model relies solely on nuclear morphology and does not incorporate spatial cell patterns or extracellular signals.

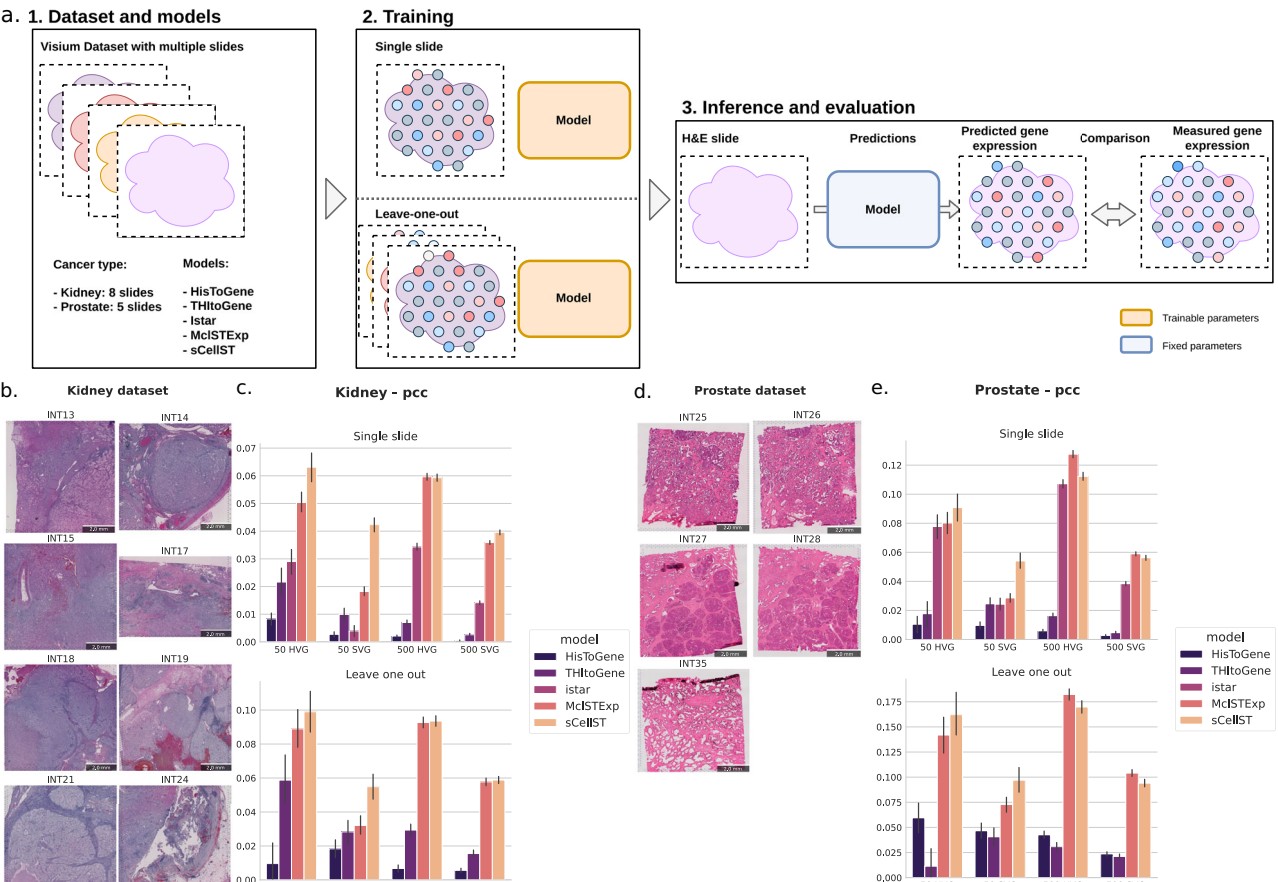

**Fig. 3 | Benchmark of sCellST. a** Overview of the benchmarking approach. The benchmark is performed on two datasets: one consisting of 8 Kidney Visium slides and the other of 5 Prostate Visium slides. sCellST is compared against four other methods (HisToGene, THItoGene, Istar, and McISTExp). We evaluated the methods under two different settings: *single-slide* training, where a model is trained on one slide and evaluated on the remaining slides, and *leave-one-out* training, where models are trained on all but one slide and evaluated on the excluded slide. The genes used for evaluation are highly variable genes (HVG) with $n$ = 50, 500 and spatially variable genes (SVG) with $n$ = 50, 500. **b, d** H&E slides from the Kidney and Prostate Visium datasets, respectively. **c, e** Benchmark results: Each bar plot represents the mean Pearson correlation coefficient (error bars correspond to the 95% confidence interval obtained via bootstrapping) for different gene sets in the Kidney and Prostate Visium datasets, respectively.

## Cell level predictions are consistent with cell type predictions from neural networks trained on manual annotations

While our analyses showed that sCellST compares favorably to state-of-the-art spot GE prediction methods, our primary objective was to predict single cell GE. For this, we compared sCellST results with state-of-the-art cell type calling methods trained on manually annotated nuclei. CellViT[21] is a widely used method for both segmentation and cell type classification, trained on more than 45000 manually annotated nuclei. The main cell types identified by CellViT are neoplastic epithelial, connective / soft-tissue cells (fibroblasts, muscle and endothelial nuclei) and inflammatory cells (methods).

To compare the coherence of our predictions with the results obtained from the DL network trained on manual annotations, we used 2 slides (Fig. 4a, b) of breast (TENX39) and ovarian (TENX65) cancer tissue from the 10X Genomics website. For each Visium slide, we independently trained and applied the sCellST pipeline, restricting the analysis to the top 1000 highly variable genes (HVGs) identified from that specific slide. We then grouped the cells according to their CellViT labels and compared the predicted gene expression scores across these groups. This approach allowed us to identify the top-expressed genes for each cell type label provided by CellViT. In Fig. 4c, d, we present the top five genes in columns for the three CellViT labels. Top differential genes were readily recognizable for the connective and inflammatory cell groups. Regarding connective cells, they encompassed genes involved in muscle contraction (MYLK) and extracellular

matrix organization (COL1A2, COL3A1, COL10A1, COL11A1, MMP2), which in turn represent specific markers of stromal cells such as muscle cells and fibroblasts, respectively. For inflammatory cells, we found classical markers of lymphocytes (IGHM), as well as other genes specific to the lymphocytic lineage (LCP1, LSP1, SRGN, POU2AF1). Due to inter-patient transcriptomic heterogeneity of epithelial tumor cells, most differentially expressed genes are expected to be patient-specific[34]. Nevertheless, the identified genes are plausible candidates for over-expression in tumoral populations: aberrant over-expression of CDCP1 is documented in breast cancer and has shown promising results as a drug target[35], and over-expression of WNT6 is documented in ovarian cancer and linked to several oncogenic mechanisms[36]. To further validate the predicted GE, we adopted a complementary approach by investigating the predicted expression of established marker genes for each of the 3 CellViT labels (Fig. 4e, f). Classical markers for epithelial tumor cells like EPCAM and E-cadherin (CDH1) showed higher levels in cells labeled as neoplastic, as did lymphocytes (PTPRC, CD3E) and fibroblasts (COL1A2, INHBA).

Next, we visually examined the coherence of CellViT and sCellST predictions. For each of the 2 Visium slides, we show a crop of the H&E image, the cell segmentation with cell types predicted by CellViT, the spots with Visium measurements and the segmented cells colored with sCellST predictions (Fig. 5). The single-cell GE predictions provide fine-grained information on the cell-type, in line with CellViT classification results, yet more detailed. In the two crops shown, sCellST allowed us

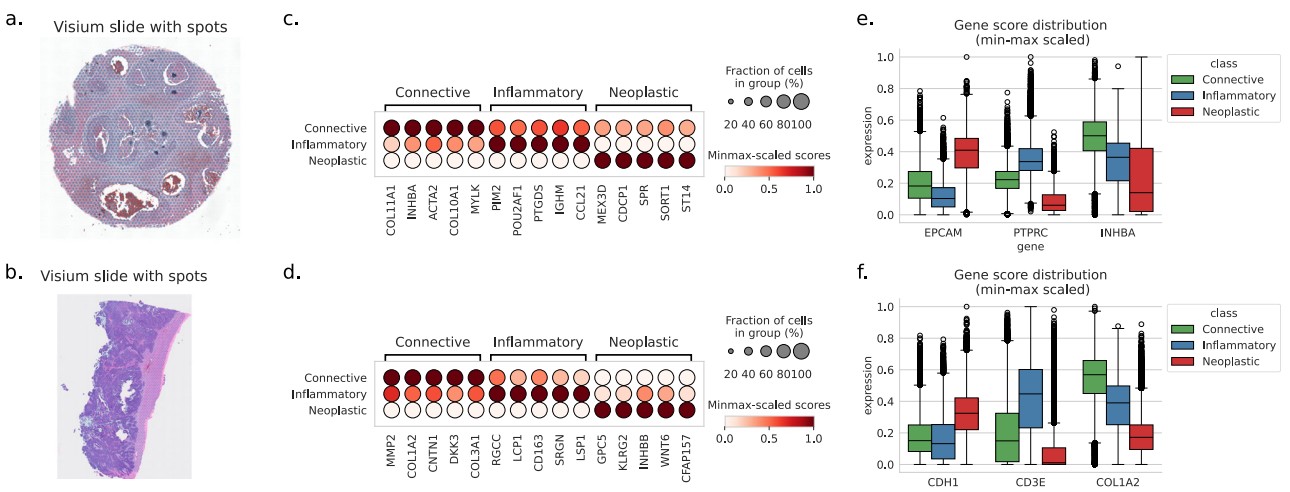

**Fig. 4 | sCellST comparison with CellViT labels.** Each row corresponds to an experiment from cancer tissues: (**a**, **c**, **e**) breast and (**b**, **d**, **f**) ovarian. **a**, **b** Visium slides used for the experiments. **c**, **d** Top differentially expressed genes when grouping cells by CellViT labels. **e**, **f** Distribution of known marker genes after min-max scaling per gene, grouped by CellViT labels and represented as boxplots (center line: median; box limits: upper and lower quartiles; whiskers: 1.5 × inter-quartile range; points: outliers). The number of cells with CellViT labels is reported in SupTable.1.

to detect the real pattern of organization of immune cells, consisting of densely populated clusters at the edges of the tumor mass. This pattern was impossible to detect with Visium resolution. In the breast cancer slide, we observed a thin layer of connective cells encapsulating the tumor that exhibited high predicted expression of INHBA. These fibroblasts could correspond to the ECM-myoCAFs[37] or the INHBA+ CAFs[38]. Lymphocyte aggregates, such as Tertiary Lymphoid Structures, play a crucial role in cancer biology[39]. Visium resolution hinders fine-grained detection and precise quantification of these structures. In the ovarian cancer slide, we observe a distinct lymphoid aggregate marked by a high density of cells with predicted expression of CD3E. Moreover, the enhanced resolution of sCellST enables the precise identification of a peritumoral pattern, at the edge of the tumor region, which entails specific consequences for the tumor microenvironment.

These analyses provide qualitative evidence of our model's relevance by showing that the distribution of predicted GE is consistent with established biological knowledge.

### sCellST is comparable to Xenium measurement in Breast Cancer

Next, we sought to compare our model's predictions, trained on a single breast cancer slide (Fig. 6a), to single-cell GE as measured by Xenium (10x Genomics). Xenium is a high-resolution imaging technology capable of measuring the expression of up to hundreds of genes at subcellular resolution. A segmentation algorithm is then applied to construct a cell-by-gene matrix. Additionally, this technology provides an H&E-stained slide. Here, we used the 9 Xenium Breast cancer samples available in the HEST database and applied sCellST to the H&E-stained slide to predict cell-level gene expression.

Comparing our model's predictions to Xenium measurements presents two main challenges. First, at the gene level, Visium and Xenium employ different capture technologies—sequencing versus imaging—which is known to introduce discrepancies in measured expression levels. Second, variations in microscopy and staining procedures pose a well-documented challenge in computational pathology[40], particularly when applying models to out-of-domain distributions. To illustrate these differences, we present galleries of randomly selected cells in SupFig.3. Furthermore, Xenium can also not be seen as an absolute ground truth, as image segmentation remains a major bottleneck for this technology.

We trained our model on the top 1000 SVG from the Visium slides and evaluated the model on the genes also measured by Xenium,

resulting in around a hundred genes evaluated depending on the Xenium slide. To address staining inconsistencies, we normalized each RGB channel based on the intensity distribution of its respective slide of origin. This approach yielded strong results without requiring more complex stain normalization techniques, as demonstrated by the distribution of Pearson correlation values in Fig. 6d, with a median Pearson Correlation Coefficient (PCC) ranging from 0.06 to 0.15. To further highlight our model's capabilities, we visualized its predictions in slide NCBI785 and TENX95 for genes associated with cancer cells: KRT8 (PCC: 0.41) and EPCAM (PCC: 0.47); and immune cells: PTPRC (PCC: 0.45) and CD3E (PCC: 0.33). In all cases, these genes enabled to highlight similar cell populations across technologies, demonstrating the robustness of our approach despite batch effects. We also provide a table with all the PCC for each gene and slide in Supplementary Data 1. However, these values should be interpreted with caution, as they are highly dependent on the training data and the specific validation slide used.

### sCellST can identify finer cell types in ovarian cancer

Automatic cell type classification in H&E data is a critical challenge in digital pathology today. The current state-of-the-art approach relies on manual cell type classifications, which presents several limitations: first, it is a formidable challenge to provide cell type annotations in sufficient number, and second, the broad categories used in manual annotation may obscure subtle morphological differences between finer cell types. In this study, we show how marker gene lists can be used to identify the morphology of more specific cell types. We focused on a publicly available human ovarian cancer slide (TENX65) from the 10X website. We trained our model to predict a list of marker genes which were obtained from an annotated single-cell dataset of ovarian tissue[41] available on the CellXGene website[42]. After training the model, we used single-cell GE predictions to define cell type scores for each cell (methods). We computed cell type scores for five cell types: fibroblasts, endothelial cells, lymphocytes, plasma cells, and fallopian tube secretory epithelial cells. We then compared sCellST scores with the broad categories provided by CellViT. Cells labeled as connective showed higher sCellST scores for fibroblasts and endothelial cells, while those labeled as inflammatory had elevated lymphocyte scores in sCellST. Lastly, cells classified as neoplastic exhibited high scores for the epithelial type in sCellST.

To understand the model's ability to distinguish finer cellular subtypes based on morphology, we subsequently generated cell image

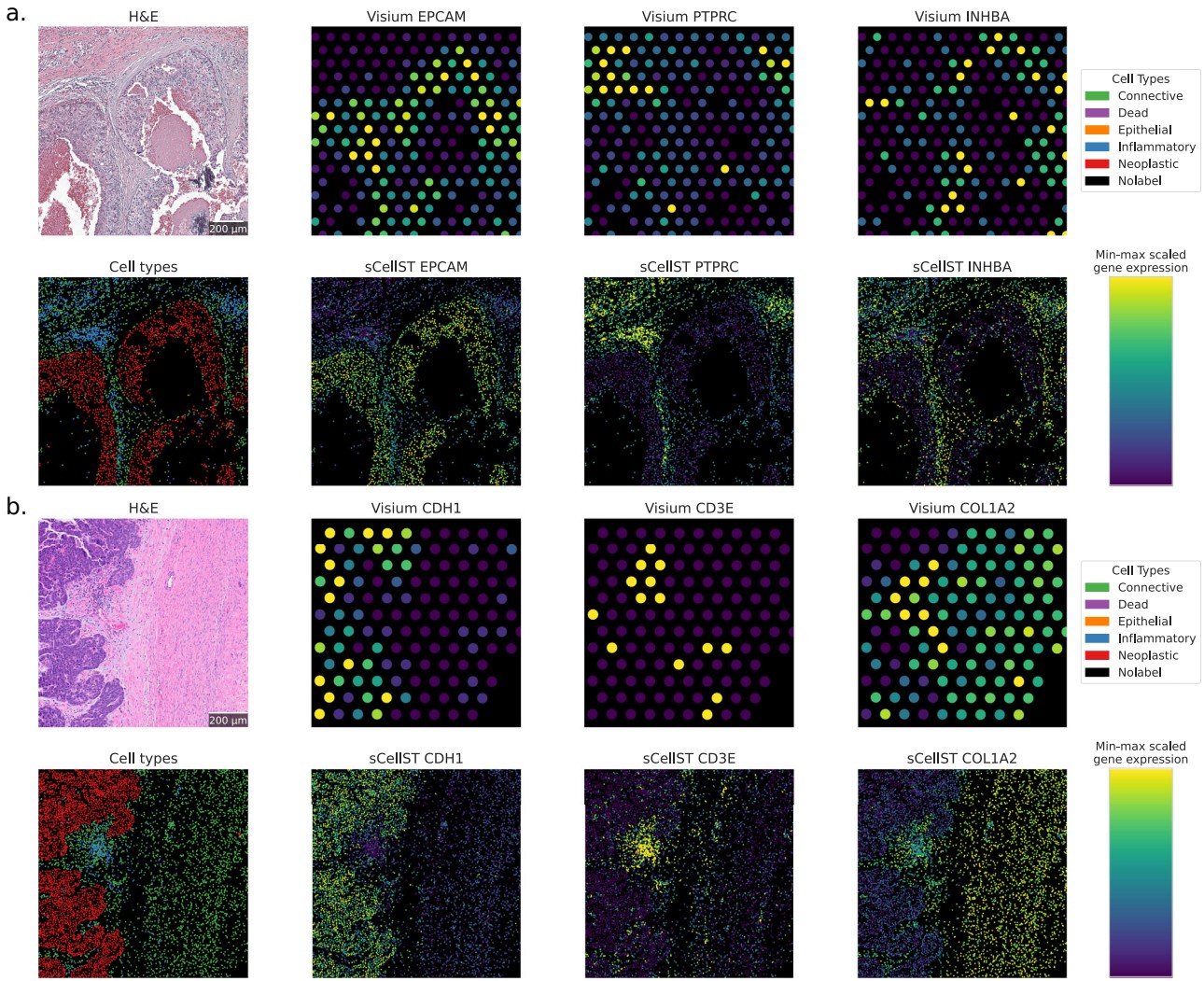

**Fig. 5 | Cell level predictions with sCellST.** Each subpanel corresponds to a Visium slide from cancer tissue: (**a**) breast and (**b**) ovarian. For all slides, the first row displays the H&E image, followed by Visium measurements in spots for three genes highlighting the presence of different cell types. The second row shows cell types predicted by CellViT, followed by sCellST gene predictions for the same genes. The minimum value in the color bar corresponds to the 0.05 quantile of each subpanel's values, and the maximum value corresponds to the 0.95 quantile.

galleries by plotting the 100 cells with the highest score for each cell type (Fig. 7c). For connective tissue cells, including fibroblasts and endothelial cells, distinct morphological characteristics could be identified from the top-scoring cell images. Although both fibroblasts and endothelial cells were spindle shaped, endothelial cells tended to be less elongated and to line a vascular space, sometimes containing red blood cells, further corroborating their identity. For inflammatory cells, the lymphocyte image gallery revealed small round cells with dark nuclei and scant cytoplasms. In contrast, plasma cells were larger and ovoid, with more abundant cytoplasms and eccentric nuclei. Additionally, we noted potential misclassification by CellViT, as cells with high plasma cell scores—labeled by CellViT as either connective or neoplastic—exhibited morphological characteristics previously associated with plasma cells (SupFig.4). Of note, while the overall spot performance of SSL- and ImageNet-encodings is similar, galleries obtained with ImageNet-encodings are less homogeneous for plasma cells and endothelial cells (SupFig.5), further underlining the importance of SSL representations for cellular phenotypes.

To assess how well this strategy generalizes to other tissue and cancer types, we tested the same method on breast cancer tissue microarray data (TENX39) using marker genes identified from scRNAseq data[43]. SupFig.6 shows the galleries extracted by sCellST.

Most morphologies correspond to what is expected for the cell type, with the notable exception of the plasmablasts class. Closer inspection of the Visium data (SupFig. 6b) revealed weak expression and diffuse spatial patterns for most of the marker genes, suggesting that the data quality for these markers was insufficient. However, when using only well expressed marker genes, the expected morphological patterns were recovered. This analysis further demonstrates that our method can accurately retrieve the correct cells based on specific cell type markers, but its performance depends on the quality of the Visium data used for training.

In conclusion, we have demonstrated that our model, when integrated with cell type marker genes, can effectively identify cells displaying distinct morphological characteristics that are not captured by state-of-the-art cell type classification models. For instance, our analyses suggest that sCellST succeeds in distinguishing fibroblasts and endothelial cells, while such subtle categorization is currently not available with state-of-the-art cell type classification models. Importantly, this analysis was conducted without reliance on manually annotated labels, yet the model successfully identified known morphological patterns solely using the predicted GE scores. Of note, this result is robust with respect to changes in the segmentation backbone (SupFig.7)

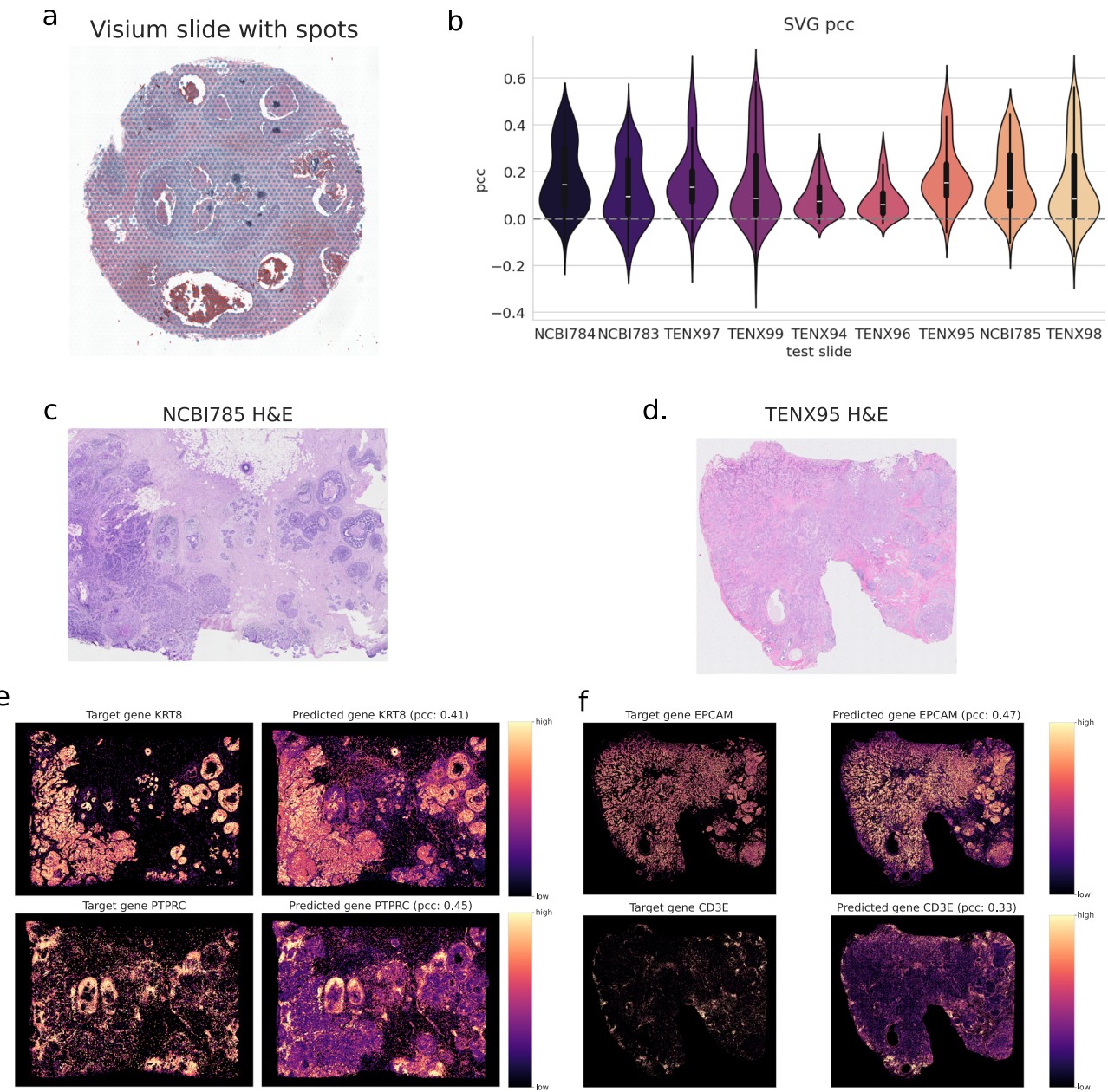

**Fig. 6 | Comparison with Xenium measurements on Breast cancer. a** Training Visium slide. **b** Distribution of Spearman correlations for predicted genes, represented with a violin plot and an embedded boxplot (center line: median; box limits: upper and lower quartiles; whiskers: 1.5 × interquartile range). **c, d** H&E slides associated with Xenium measurements. **e, f** Comparison between measured genes (left sub-column) and sCellST predictions (right sub-column). The minimum value in the color bar corresponds to the 0.01 quantile of each subpanel's values, and the maximum value corresponds to the 0.99 quantile.

## Discussion

We presented sCellST, a method for predicting single-cell and spatially resolved GE from H&E images based on a weakly supervised learning framework. Unlike other approaches, sCellST generates a detailed spatial map of cell-type-specific expression patterns from H&E data, providing a more granular understanding of GE at the single-cell level.

Although not originally designed for this task, we demonstrated that sCellST achieves performance that is comparable to state-of-the-art models for spot-level gene expression prediction. We also showed that sCellST accurately predicts gene expression with high correlation to Xenium measurements, despite significant out-of-domain variations at both the image and GE level. Moreover, sCellST produces results on par with leading methods trained on tens of thousands of manually annotated nuclei for broad cell type classification. Notably, unlike these methods, sCellST can distinguish finer-grained cell types by leveraging subtle morphological differences that would be challenging or impossible to capture through manual annotation.

For these reasons, we believe sCellST has the potential to drive several important developments. First, it enables large-scale studies of the relationship between nuclear morphologies and GE, facilitating the identification of cell type-specific morphologies. Second, sCellST introduces a novel annotation strategy for single-cell computational pathology (SCCP), which currently depends heavily on extensive manual annotations. For example, Diao et al.[44] trained cell classification models on 1.4 million manually annotated cells by certified pathologists. sCellST offers an efficient way to generate large-scale single-cell annotations with minimal manual input, while also distinguishing more fine-grained cell types. As such, sCellST could significantly impact SCCP. Finally, the ability to dissect cell types from H&E images opens up unprecedented opportunities to reanalyze existing H&E cohorts for predicting outcomes

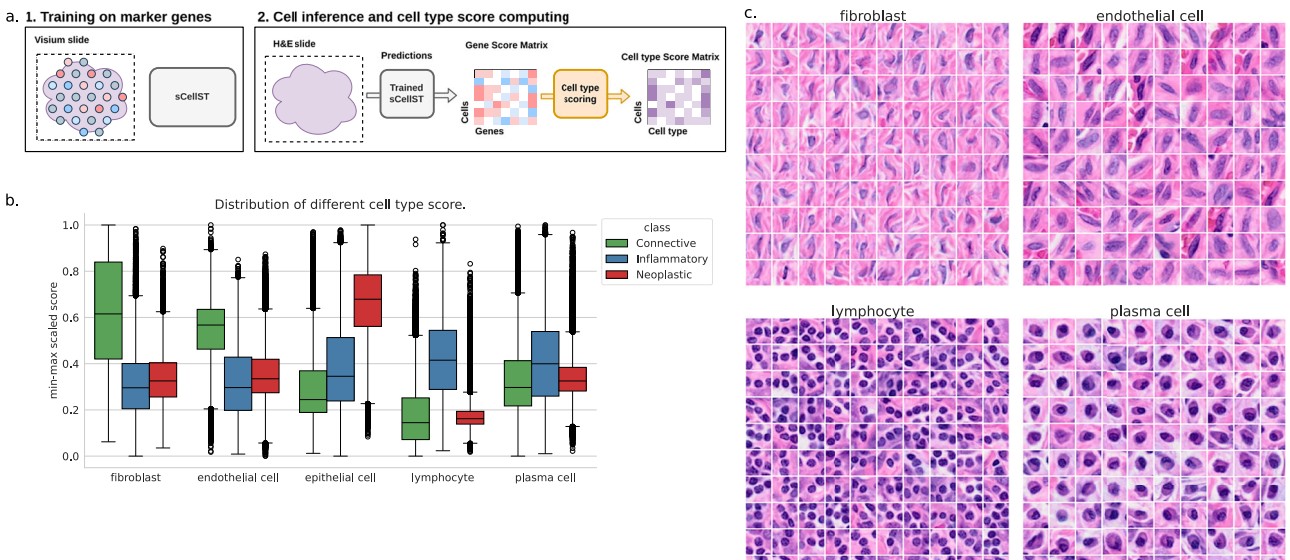

**Fig. 7 | sCellST discovers cell type morphological features. a** Schematic of the approach. First, sCellST is trained on a set of marker genes obtained from prior knowledge. Then, a scoring function is used to compute cell type scores for each cell. **b** Distribution of scores after min-max scaling per score for several cell types, grouped by CellViT labels and represented as boxplots (center line: median; box limits: upper and lower quartiles; whiskers: 1.5 × interquartile range; points: outliers). The number of cells for CellViT labels is reported in Supplementary Material (SupTable.1) **c** Image galleries showing the highest-scoring images for each cell type.

and treatment responses. Although ST is a powerful technique, it is unlikely to be applied to large retrospective cohorts in the near future due to its high cost and limited tissue availability. sCellST offers the possibility to create high-resolution virtual ST. While virtual ST may not fully match the quality of direct measurements, it is still expected to provide valuable and exciting insights.

The proposed method is not free of limitations. First, sCellST relies on nucleus segmentation, making it vulnerable to segmentation errors. While the effect of different segmentation methods seems to be minor, segmentation errors induced by differences in image acquisition (staining and scanning) can affect method performance. These technical variations also affect the generalizability of the cell image embeddings, ultimately limiting the broader use of our model without retraining. Future research may tackle this challenge by creating cell image representations that are robust across domains; however, this remains an active area of investigation. Notably, even so-called *foundation models* operating at the patch level have not yet fully overcome these issues[45]. Added to this, ST data is still scarce and consequently, the small size of training datasets negatively impacts predictive performance and generalization. Of note, our method requires high-resolution FFPE data, as the prediction relies on relatively small image patches of individual cells. Consequently, any compromise in image quality will inevitably degrade prediction performance. While several initiatives have begun collecting large-scale datasets[33,46], the availability of FFPE Visium slides and access to their corresponding high-resolution images remain limited. This scarcity currently hinders large-scale training—a prerequisite for model transferability without retraining. At the same time, spatial resolution of sequence-based ST is increasing[47]. In Visium HD, bins are arranged on a 2 $\mu m$ rectangular grid. This higher resolution could enhance GE prediction but requires custom training techniques, as bins are typically grouped into 8 $\mu m$ sizes and do not align with individual cells.

Overall, we believe that our approach can serve as a pioneering method for predicting GE from cell morphology in H&E images. With the scaling of ST dataset sizes, it has the potential to predict GE on large cohorts of H&E images, facilitating novel biological discoveries.

## Methods

### Notations and abbreviations

- $S$: number of spots in a Visium slide
- $G$: number of genes
- $f_\theta$: a neural network parametrized by a set of weights $\theta$
- $\mathbf{x} \in \mathbb{R}^{h \times w \times 3}$: a RGB cell image of size $h \times w$
- $\mathbf{h} \in \mathbb{R}^d$: a cell embedding vector of dimension $d$
- $\mathbf{y} \in \mathbb{N}^G$: a raw vector of gene expression and $Y \in \mathbb{N}^{S \times G}$ the spot GE matrix
- $\mathbf{y}^p \in \mathbb{R}^G$: a preprocessed vector of GE and $Y^p \in \mathbb{R}^{S \times G}$: the pre-processed spot gene expression matrix
- We used the notation $\hat{y}$ to denote predictions, in this case of a raw GE vector.
- $p_{NB}(.\,; a, b)$: density function of a negative binomial distribution parametrized by the parameters $a$ and $b$ corresponding to either $\mu$ (mean) and $\theta$ (inverse overdispersion) used in ref. 48 or total counts and probability of success parametrisation.
- $\mathcal{L}_{mse/nll}$: loss function (Mean Squared Error or Negative Log Likelihood)

### Spatial transcriptomics

We based our approach on Visium technology because it provides both a spatially resolved transcriptomic profile and a corresponding H&E image. Visium is part of the spot-based ST family, capturing mRNA using spatial barcodes within defined spots that typically contain 10–20 cells. The captured mRNA is then sequenced using next-generation sequencing (NGS) technology. A key advantage of this method, compared to image-based ST, is its ability to capture the entire transcriptome, though at a lower spatial resolution. In our study, we utilized Formalin-Fixed Paraffin-Embedded (FFPE) ST datasets, as FFPE preserves cellular morphology more effectively than fresh frozen tissue samples. Additionally, we analysed H&E images at the highest available resolutions, ranging from 0.2 to 0.5 $\mu m$ per pixel, depending on the specific slide.

For the validation, we used Xenium which is a high-resolution imaging technology capable of measuring the expression of up to hundreds of genes at subcellular resolution.

## ST preprocessing

For each Visium slide, we filtered genes with less than 200 counts and those that were detected in less than 10% of the training spots. Furthermore, we filtered spots for which no cells were detected and those with less than 20 counts. These filtering steps exclude genes with very low expression on the slide and therefore unlikely to be predictable, as well as uninformative spots. In our experiments, we used either custom gene lists (based on known marker genes informative of specific cell types), Highly Variable Genes (HVGs), or Spatially Variable Genes (SVGs). For the simulation experiments, HVGs were selected from the scRNA-seq atlas to ensure a consistent gene list across all scenarios. In the benchmarking studies, HVGs were identified on all slides from a given dataset using the Scanpy[49] function *highly_variable_genes*, with *flavor='seurat_v3'* and *batch_key=slide_idx* to avoid selecting slide-specific genes. For SVGs, we computed Moran's Index on each slide and ranked genes based on the geometric mean of their scores across all slides. Finally, for the remaining experiments, we selected HVGs computed directly on the Visium slides.

For Xenium, we filtered genes with less than 5 counts. GE values were also log-transformed.

## Cell segmentation from whole slide images

For the segmentation step, we utilized a publicly available pre-trained network called CellViT[21], which simultaneously performs cell segmentation and classification. For each segmented nucleus, we extracted a 12$\mu$m × 12$\mu$m (typical cell size) image centered on the cell's segmentation center coordinates, which was resized to a 48 × 48 pixel images. Based on the spatial coordinates of the spots, cells, and the spot radius, we linked each cell to its corresponding spot. This kind of models is usually not applied to the whole slide for memory reasons but to tiles (small patches) which cover the tissue. The algorithm used in this study, CellViT, predicts six distinct classes: neoplastic epithelial cells, connective/soft-tissue cells (including fibroblasts, muscle, and endothelial nuclei), inflammatory cells, non-neoplastic epithelial cells, dead cells, and unlabeled cells. In our analysis of the H&E slides, we observed that the majority of cells were classified into one of the first three categories (see SupTable.1). Therefore, we restricted our analysis to these three primary classes.

## Image embedding

Given the limited amount of data available for training the GE predictor, we employed strategies to obtain image embeddings independently from this prediction task. Specifically, we explored two approaches: Transfer Learning[50] from ImageNet classification[51] and SSL. In both cases, we used a ResNet-50[52] backbone as encoder. In both scenarios, NN are utilized to map input images to generic representations. For transfer learning, these representations are obtained by training the NN on entirely different data and entirely different tasks (such as classification of natural images). SSL is optimized for the same type of data (in our case histopathology data), but is trained on so-called pretext tasks, that do not require any annotation or external ground-truth. Among the various available SSL methods, we opted to use MoCo v3, a contrastive learning framework. Briefly, this approach generates two different views of each input image (i.e., transformed using specific augmentations) and is optimized to pull together the corresponding representations for views originating from the same image, while pushing apart those that originate from different images. Choosing the right augmentations for image transformation is key to obtain good representations and can heavily influence the performances of downstream tasks[53]. Since H&E images are very different from natural images, we changed the augmentations and worked with different parameters for color jittering. We also added two other transformations: random erasing and random rotations. For the latter, we extracted larger cell images as to avoid rotation induced artifacts

when finally cropping images to 48 × 48 pixels. We trained one SSL network per cancer type. To train it, we used all cells from the Visium slides. This scenario allows to reduce computations for our experiments and reflect a setting where we would use all the image data available to train the SSL network including Visium slides and H&E slides on which to infer gene expression. There is no information leakage since the SSL network does not see any gene expression value during training. In the simulation experiment, we also used one-hot cell type encodings as cell image representations. This corresponds to a vector that is one for the correct cell type, and 0 otherwise.

## Multiple instance learning (MIL) for spatial transcriptomics

Unlike the classical supervised learning framework, Multiple Instance Learning is designed for learning in the case when a single label $y \in \mathcal{Y}$ is available for a set of instances $\{x_1, x_2, ..., x_k\}$, referred to as a bag where $x_i \in \mathcal{X}$ for $i = 1, ..., k$. where $k$ is the number of instances in the bag. Although each instance in the bag has an underlying label, these individual labels remain inaccessible during training.

In the instance-level approach of MIL algorithms, the objective is to learn an instance-level predictor $f_\theta$ which assigns a score to each instance. These instance-level scores are then aggregated using a function $g$ to generate a score for the entire bag. As the order of instances in the bag is irrelevant, $g$ must be permutation invariant. Common choices for $g$ include *mean* or *max* operations, depending on the specific task.

As every operation presented above can be chosen differentiable, the instance-level predictor $f_\theta$ is often parametrized using neural networks, which has proven effective, particularly when working with image data. Such approaches are frequently applied to tasks in computational pathology in the general formulation of MIL[54].

The instance-based MIL framework is particularly well-suited for spot-based ST, as each spot $s$ within a slide can be seen as a set of cells which represent the instances. For each spot index $s \in \{1, ..., S\}$, we have a target GE vector $\mathbf{y}_s \in \mathbb{R}^G$ and a set of images $\{\mathbf{x}_1, \mathbf{x}_2, ..., \mathbf{x}_{k_s}\}$ derived from the detection algorithm where $k_s$ represents the number of cells in the spot $s$.

Cell embeddings $\{\mathbf{h}_1, \mathbf{h}_2, ..., \mathbf{h}_{k_s}\}$ are produced using a pretrained embedding model $f_\phi$.

$$\mathbf{h}_i = f_\phi(\mathbf{x}_i), for\ i = 1, ..., k_s$$

A feed-forward neural network, $f_\theta$, is then trained to predict a vector of GE scores based on the cell embeddings. The goal was to develop an algorithm which could predict biologically relevant GE scores at the single-cell level. In order to ensure positiveness of the single cell GE scores, we used a softplus function as final activation function. Finally, a mean aggregation function was applied to simulate the measurement process and derive a GE estimate for comparison with the measured spot-level GE.

$$\widehat{\mathbf{y}}_s = \frac{1}{S} \sum_{i=1,...,S} f_\theta(\mathbf{h}_i)$$

## Model loss function

Two types of objective functions were considered for training the GE predictor.

The first utilized a mean squared error (MSE) loss with preprocessed GE data. In this approach, GE values were normalized by library size and log-transformed, following standard practices in spatial transcriptomic and single-cell analyses:

$$y_j^p = \ln(1 + s\frac{y_j}{\sum_j y_j}) for\ j = 1, ..., G$$

$s$ is a normalization constant set to 10000. In this case, the loss function becomes:

$$\mathcal{L}_{mse}(\widehat{\mathbf{Y}}, \mathbf{Y}) = \frac{1}{SG} \sum_{i=1}^{S} \sum_{j=1}^{G} (\widehat{Y}_{ij}^{p} - Y_{ij}^{p})^2$$

To make sure that all genes contributes to the loss, we additionally scaled each gene to a similar rage using a robust scaler from scikit-learn[55]. In the case of multiple slide training, we applied this scaling slide-by-slide.

The second objective function was based on minimizing the negative log-likelihood on raw counts by modeling GE data using a negative binomial distribution. In this formulation, predictions were made for either the mean or the total count parameter of the distribution. The other parameter $\alpha_j$, chosen to be gene specific, is learned during the training. In both cases, the observed library size divided by the median library size was used as a scaling factor $l_i$ to avoid capturing this information in the single-cell scores. In this case the loss function becomes:

$$\mathcal{L}_{nll}(\widehat{\mathbf{Y}}, \mathbf{Y}) = -\frac{1}{S} \sum_{i=1}^{S} \sum_{j=1}^{G} \ln(p_{NB}(Y_{ij}; l_i \widehat{Y}_{ij}, \alpha_j))$$

Both approaches were evaluated through simulations; no significant advantage was observed with the negative binomial formulation, as illustrated in the simulation experiments (SupFig.8). Thus, the MSE-based approach was retained for subsequent analyses.

### Model training and implementation

All models were implemented and trained with Pytorch[56]. For gene expression predictors, we used a fully connected network with three hidden layers of dimension 256 each. We used the AdamW[57] optimizer with a learning rate of $10^{-4}$ and a batch size of 128. We held-out 20% of the training set as a validation set that we used to select the best model using an early stopping approach with a patience of 20. For experiments using Xenium slides and marker genes, we found that applying a robust scaler to normalize gene expression values to the [0, 1] range improved the overall quality of the results.

### Model evaluation

To evaluate model performance, we focused on correlation-based metrics, namely the *Pearson Correlation Coefficient (PCC)* and the *Spearman Correlation Coefficient (SCC)*. Given two vectors of real numbers,

$$\mathbf{a} = (a_1, a_2, \ldots, a_n) \in \mathbb{R}^n \text{ and } \mathbf{b} = (b_1, b_2, \ldots, b_n) \in \mathbb{R}^n,$$

these metrics are defined as follows:

$$
\begin{aligned}
PCC(\mathbf{a}, \mathbf{b}) &= \frac{\sum_{i=1}^{n}(a_i - \overline{\mathbf{a}})(b_i - \overline{\mathbf{b}})}{\sqrt{\sum_{i=1}^{n}(a_i - \overline{\mathbf{a}})^2}\sqrt{\sum_{i=1}^{n}(b_i - \overline{\mathbf{b}})^2}}, \\
SCC(\mathbf{a}, \mathbf{b}) &= PCC(rank(\mathbf{a}), rank(\mathbf{b})),
\end{aligned}
$$

where $\overline{\mathbf{a}}$ and $\overline{\mathbf{b}}$ denote the sample means of $\mathbf{a}$ and $\mathbf{b}$, and rank($\mathbf{a}$), rank($\mathbf{b}$) represent the ranks of the elements in each vector.

The rationale for selecting these metrics is that PCC captures linear relationships, while SCC captures monotonic (not necessarily linear) dependencies. In particular, PCC remains high even when the relationship is linear with a slope different from 1, which is useful in gene expression prediction where batch effects can shift absolute expression values across slides.

For completeness, we also report additional regression metrics for the leave one out training setting on the Prostate dataset (SupFig.9), including the *Root Mean Squared Error (rMSE)* and the *Mean Absolute Error (MAE)*, defined as:

$$
\begin{aligned}
rMSE(\mathbf{a}, \mathbf{b}) &= \sqrt{\frac{1}{n} \sum_{i=1}^{n} (a_i - b_i)^2}, \\
MAE(\mathbf{a}, \mathbf{b}) &= \frac{1}{n} \sum_{i=1}^{n} |a_i - b_i|.
\end{aligned}
$$

While these metrics reflect similar trends to PCC and SCC, they are more sensitive to scale and harder to interpret across genes or datasets. Therefore, we primarily report PCC and SCC in the main text.

### Simulations

We used a scRNA-seq dataset from CellXGene of ovarian cancer cells and an H&E slide from a Visium slide to perform our simulation study. After training the SSL model on cells extracted from the H&E, we clustered the image embeddings with a k-means algorithm (k=6) to identify distinct morphological clusters. We kept only the closest 2000 cells to each cluster centroid in order to have strong morphological differences between clusters. To assign gene expression vectors to cell images, we arbitrarily matched six morphological clusters with six annotated cell-type clusters from the scRNA-seq dataset. While this matching does not reflect true biological relationships, it ensures a strong correspondence between morphological and transcriptomic features for evaluation purposes. GE was then assigned to each cell image based on three scenarios to evaluate the model's performance under different levels of association between GE and cell morphology:

1. Centroid scenario - perfect link between cell morphology and GE: The mean GE of the corresponding scRNA-seq cluster was assigned to each cell image (SupFig.10.a)
2. Random scenario - no link between cell morphology and GE: Each cell image was assigned a random GE vector from the scRNA-seq dataset (SupFig.10.b)
3. Cell scenario - partial link between cell morphology and GE: Each cell image was assigned the GE vector of a cell from the corresponding scRNA cluster (SupFig.10.c)

It is important to note that, in this simulation only, the SSL embeddings carry no additional information beyond the morphological clusters−unlike in the other experiments presented in this study. This explains why one-hot-encoded features perform better in Fig. 2e. For each scenario, we generated 5000 spots, with each spot containing 20 cells for both training and testing sets. These numbers were chosen to reflect the setting typically observed in Visium ST slides. The model was trained using the top 1000 highly variable genes (HVGs) selected on the single cell dataset. Among these 1000 HVGs, we also used marker genes (MG) corresponding to each annotated cluster resulting in 51 genes. To evaluate model performances, we computed the Pearson and Spearman correlations, at both the spot and cell levels, on log-normalized GE vectors for all models.

### Cell prediction downstream analysis

Following cell GE prediction using a trained sCellST model, we utilized standard tools in Scanpy for downstream analysis. Specifically, we performed differential expression analysis using the *rank_genes_groups* function of Scanpy with a t-test to rank genes after grouping cells based on CellViT labels. The number of cells for each CellViT are reported in SupTable.1. For the cell type analysis with marker genes, we first identified cell type marker genes with a reference single-cell dataset[41] from CellXGene. We selected only cells originating from the left and right ovary. We applied a t-test with overestimated variance (two-sided) to identify 20 marker genes per annotated cell type cluster (see SupTable.3, 4). From the scRNAseq dataset, we identified marker

genes for cell type clusters, of which 130 remained after filtering. B and T cells were merged to form a lymphocyte group, and we excluded mast cells, monocytes, and dendritic cells from the analysis because of the difficulty to unambiguously recognize them in H&E images and thus to validate the morphology galleries produced by our method. We normalized the predictions such that each cell has the same amount of expression and then scaled the expression values of each gene. We subsequently used the *score_genes* function of Scanpy to compute signature scores for cell type marker genes (see Sup. Table 1). For each cell $i$, the scoring process involved two lists: a marker gene list $G_m$ and a control gene list $G_c$. The score of a cell for the marker gene list $G_m$ was calculated as follows:

$$s_m = \frac{1}{G_m}\sum_{i \in G_m}\widehat{y}_i - \frac{1}{G_c}\sum_{j \in G_c}\widehat{y}_j$$

### GE predictors

We compared our approach to four other state-of-the-art methods for GE prediction from H&E data. We made some modifications in the original code to adapt them to the training data presented in this study and to make fair comparisons but otherwise we kept all the hyper-parameters defined in the original codes.

- HisToGene is based on a Visual Transformer architecture which take as input images of spots alongside binned spatial coordinates. As it has been originally implemented for Spatial Transcriptomics data (previous version of Visium) which contain fewer and larger spots per slide, we increased the number of positional encodings from 64 to 128 to enable error-free model training.
- THItoGene is also based on a Visual Transformer but also introduces capsule network and graph attention layer to improve the model. Similarly, we increased the number of position in the positional embedding to 128 to adapt the Visium slides in this study.
- Istar is also a weakly supervised approach. It takes as input small patches embedded with a pretrained network before making a prediction for each patch and comparing it to the spot measurement to train the model. We included it only in the single slide benchmark experiment because the provided code was not adapted to the training of multiple slides at the same time. Istar trains five neural networks and aggregates their predictions to obtain the final output, a technique known as *ensembling* in machine learning and statistics. For this study, we reduced the number of trained models from five to one to ensure fair comparison with other approaches which could also benefit from ensembling. Indeed, ensembling can be applied to every method and usually enhances performance.
- MclSTExp is a contrastive based approach. This kind of algorithm does not predict directly gene expression but instead try to align representation of both images and gene expression. Once trained, the model can be used to infer the gene expression of an unseen image by using a weighted average of the gene expression associated to the closest image embeddings in the training set.

### Reporting summary

Further information on research design is available in the Nature Portfolio Reporting Summary linked to this article.

## Data availability

We accessed the spatial transcriptomic data used in this study within the HEST database[33] hosted at https://huggingface.co/datasets/MahmoodLab/hest with the provided cell segmentation. For each cancer type, we include the slide ID from HEST along with links to the original publication or source website. We excluded slides which were not preserved with FFPE and where the H&E staining quality was insufficient for the cell segmentation algorithm to perform effectively. Visium slides HEST ids: • Prostate: INT25, INT26, INT27, INT28, INT35 • Kidney[32]: INT13, INT14, INT15, INT17, INT18, INT19, INT21, INT24 • Breast: TENX39• Ovary: TENX65. Xenium slides HEST ids (with publication of raw dataset links): • NCBI783, NCBI784, NCBI785[4]• TENX94, TENX95 https://www.10xgenomics.com/datasets/ffpe-human-breast-with-pre-designed-panel-1-standard, • TENX96, TENX97 https://www.10xgenomics.com/datasets/ffpe-human-breast-with-custom-add-on-panel-1-standard, • TENX98, TENX99 https://www.10xgenomics.com/datasets/ffpe-human-breast-using-the-entire-sample-area-1-standard, single cell RNA dataset: • Ovary[41]: https://datasets.cellxgene.cziscience.com/73fbcec3-f602-4e13-a400-a76ff91c7488.h5ad• Breast[43]: https://datasets.cellxgene.cziscience.com/fabd4946-3f41-459c-ba79-188749a8baa4.h5ad. Source data are provided with this paper.

## Code availability

The code used to develop the model is publicly available and has been deposited in sCellST at (https://github.com/loicchadoutaud/sCellST[58]) under CC-BY 4.0 license. The code used to perform the analyses and generate results in this study is publicly available and has been deposited in sCellST_reproducibility at (https://github.com/loicchadoutaud/sCellST_reproducibility.git).

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

## Acknowledgements

We would like to thank Raphaël Bourgade, Lucie Gaspard-Boulinc, Nicolas Captier, Nicolas Servant and Loredana Martignetti for helpful discussions. This work was funded by the French government under the management of Agence Nationale de la Recherche as part of the "Investissements d'avenir" program, reference ANR-19-P3IA-0001 (PRAIRIE 3IA Institute; grants to TW and EB) and ANR-23-IACL-0008 (PRAIRIE-PSAI; grants to TW and EB). Furthermore, this work was supported by ITMO Cancer (20CM107-00; grant to TW). Furthermore, this work was supported by a government grant managed by the Agence Nationale de la Recherche under the France 2030 program, with the reference numbers ANR-24-EXCI-0001, ANR-24-EXCI-0002, ANR-24-EXCI-0003, ANR-24-EXCI-0004, ANR-24-EXCI-0005 (grants to TW and EB).

## Author contributions

L.C., M.L., E.B., and T.W. designed and planned the study. L.C., M.L., and J.O. developed the tool. L.C., D.H., and J.F. performed the analysis. E.B. and T.W. supervised the study. L.C., D.H., J.F., E.B., and T.W. wrote the manuscript. All authors reviewed and/or edited the manuscript prior to submission.

## Competing interests

The authors declare no competing interests.
