## [Transparent Peer Review file · Nature Communications]

sCellST Predicts Single-Cell Gene Expression from H&E Images

Corresponding Author: Professor Thomas Walter

Version 1:

Reviewer comments:

Reviewer #1

(Remarks to the Author)

The study presents an innovative computational pathology method (sCellST) with potential for predicting single-cell gene expression from H&E images. It contributes valuable insights into the link between morphology and transcriptomics, with applications in large-scale tissue studies. However, I recommend to address the following concerns to enhance this study.

- 1) The introduction is well-motivated, but it repeats background information on spatial transcriptomics in multiple places. The discussion of spatial transcriptomics (ST) vs. image-based ST methods could be more concise.
- 2) The MIL framework is well explained, but the notations are somewhat dense and scattered across sections. A single table summarizing key mathematical terms (e.g., GE prediction functions, loss functions, MIL assumptions) would improve clarity. Self-supervised learning (SSL) details should be clearly separated from MIL in a distinct subsection for easier comprehension.
- 3) Some figures lack clear p-values or confidence intervals, which should be added for robustness. Also, figure legends should clearly explain the key comparisons (e.g., benchmarks across models, cell-type predictions).
- 4) The study primarily reports Pearson/Spearman correlations for performance evaluation. However, additional metrics such as Mean Absolute Error (MAE), Root Mean Square Error (RMSE), or AUROC could further validate the model.
- 5) The study lacks an explicit analysis of inter-slide variability, how well does the model generalize across datasets with different staining protocols?
- 6) It would be helpful to include an ablation study to assess the contribution of different components (e.g., how much does SSL improve performance over a simple MIL baseline?).
- 7) The manuscript mentions using Highly Variable Genes (HVGs) and Spatially Variable Genes (SVGs) but does not clearly justify the choice of cutoff thresholds. A sensitivity analysis on different gene selection criteria (e.g., top 1000 vs. top 500 genes) would strengthen the findings.
- 8) The study demonstrates that sCellST can predict known cell types, but it does not explore novel biological insights gained from these predictions. Could sCellST uncover previously unrecognized morphological-genetic associations in tumors?

(Remarks on code availability)

The codes are prepared well for users.

Reviewer #2

(Remarks to the Author)

* What are the noteworthy results? *

Predicting gene expression directly from histology is an active research area, and the authors position sCellST against two broad classes of existing approaches. Within this landscape, sCellST offers a unique contribution by delivering single-cell

resolution predictions via a MIL paradigm, indicating the novelty of this work.

* Will the work be of significance to the field and related fields? How does it compare to the established literature? If the work is not original, please provide relevant references. *

Although there are similar ideas being published previously (e.g., STAnD [2]), but the authors of this manuscript have conducted a comprehensive research focusing on establishing a pipeline with more advanced approaches for predicting gene expression using single cell H&E images.

* Does the work support the conclusions and claims, or is additional evidence needed? *

Yes, the work supports the conclusions and claims

* Are there any flaws in the data analysis, interpretation and conclusions? Do these prohibit publication or require revision? *

To the reviewer's point of view, the correlation between predicted gene expression and actual gene expression is a critical information for evaluating the performance of the proposed pipeline. The authors are suggested to provide this comparison.

* Is the methodology sound? Does the work meet the expected standards in your field? *

Yes

* Is there enough detail provided in the methods for the work to be reproduced? *

Yes

Reviewer's comment:

The authors propose sCellST, a pipeline that predicts gene expression at single-cell resolution from H&E-stained tissue images by training on paired spatial transcriptomics (ST) data. The methodology is clearly structured into three main components: (1) Cell Detection and Segmentation, (2) Self-Supervised Feature Extraction, and (3) Multiple Instance Learning (MIL) gene expression prediction as shown in Fig.1 of the manuscript. Overall, the approach is logically sound and well-described, with each step justified by prior work or preliminary experiments.

The authors anticipate the main challenges (domain shift and segmentation/classification errors) and discuss them. The model's performance on single-cell data (Xenium) and across multiple tissues indicates its performance and can be adapted relatively easily (as the result has suggested the overfit is contaminated). With more training data and perhaps minor methodological tweaks (domain augmentation), sCellST could likely be deployed widely.

Below are some points that the reviewer

1. In the article, the authors stated that "CellViT, predicts six distinct classes: neoplastic epithelial cells, connective/soft-tissue cells (including fibroblasts, muscle, and endothelial nuclei), inflammatory cells, non-neoplastic epithelial cells, dead cells, and unlabelled cells. In our analysis of the H&E slides, we observed that the majority of cells were classified into one of the first three categories (see SupTable.1). Therefore, we restricted our analysis to these three primary classes." I agree that cancer cells, connective cells and inflammatory cells are possibly the easiest 3 cell categories which is able to be classified at single cell level, while other cell types could be more challenging (although pathologists may be able to classify cell types based on bigger picture rather than using nuclei images). The author may want to evaluate the impacts of using a different cell detection / classification approach? E.g., the mentioned hover-net.

2. Although the authors have discussed the prediction performance at bag and instance level (via the perspective of MIL) in Fig 6. However, biologists would be more interested on the correlation between the predicted gene "expression" v.s. the actual gene expression. E.g., computing the "pseudo spot" similar to [2] and the spearman correlation. This approach will be able to provide not only the gene list of high correlations (as shown in Fig. 7) but also provide the level of correlation in the ranked gene list.

3. Single-cell level gene expression prediction represents a great challenge. Various researchers have tackled this challenge, e.g., XFuse [1], aims to predict the whole slide gene expression at pixel level. STAnD [2] predicting single-cell gene expression based on H&E using MIL (which is similar to the idea proposed in this manuscript).

4. Some references are missing, e.g., Transfer learning, ImageNet, Self-supervised learning and, ResNet-50 (page 23, line 1055).

5. There are some duplicated acronyms, e.g., SSL in page 5, line 228 and SSL in page 23 line 1055.

6. The authors are suggested to revise the manuscript more carefully. Although the reviewer didn't have spotted major error in the manuscript, but there are some errors, misspelling, etc., have to be addressed., including but not limited to line 1273, page 28, CellViT (is it CellViT with the lower case "i")

REFERENCES

[1] Bergensträhle, L., He, B., Bergensträhle, J. et al. Super-resolved spatial transcriptomics by deep data fusion. *Nat Biotechnol* 40, 476–479 (2022). <https://doi.org/10.1038/s41587-021-01075-3>

[2] Huang C. H., Park Y.; Pang, J., Bienkowska, J. R., Single-cell gene expression prediction using H&E images based on spatial transcriptomics, *Proceedings Volume 12471, Medical Imaging 2023: Digital and Computational Pathology*; 1247105 (2023) <https://doi.org/10.1117/12.2654294>

(Remarks on code availability)

Reviewer #3

(Remarks to the Author)

Summary

The paper proposes a new method for predicting gene expressions from histology images. The method is based on multiple instance learning on a bag of features derived from single cell images. The images come from a cell segmentation model and the features are obtained from a self-supervised feature extractor. Despite being trained on spot expressions, the model

appears to be able to predict single-cell expressions, which is novel. Although the method has been evaluated on several tasks and both simulated and real datasets, the results presented are not sufficient to validate the strong claims made throughout the paper, see below for detailed comments.

Analysis

The prediction of single cell expressions is an interesting and valuable topic. Since there is no ground-truth available for the single cell expressions paired with histology images in the Visium samples, the paper suggests using the CellViT segmentation labels and their corresponding annotated scRNA-seq expressions together to get a broad perspective of the expressed genes for each cell in histology. While this is a well thought approach several questions arise in various experiments which are crucial to support the credibility of these claims. We have listed them in the following points: Section 2.2 - The paper states it can predict the single cell level gene-expression in simulated data. This is shown with an example from an ovarian cancer visium slide and an annotated single cell reference taken from CellXGene. However the details on the biological reasoning of how the image and single cell expression clusters are matched are missing. For example, in the 312-314 centroid scenario, it's unclear how the clusters were matched (i.e what is the corresponding scRNA-seq cluster?). Similarly 315-317 what is the corresponding morphological cluster and how is it decided to match the image and scRNA clusters? Further details need to be added to line 1228-1231 to support this experiment. Additionally, the simulation experiment has been performed only on a single visium slide of ovarian cancer which limits the heterogeneity observed across visium samples and spatial-omics in general. The experiments need to be done on multiple samples to support the claim.

Line 326 - 355 shows high mean correlation between the ground-truth spot and predicted aggregate of cell expressions for the top 1000 highly variable genes. This experiment is done to show that the MIL approach can recover GE vectors from cell images. It's important to note, it is quite ambiguous in this section as well as in the methods section whether the HVGs are computed on the Visium samples or single cell atlas. This is crucial since it's not necessary that the spot and cellular level HVGs are identical.

While pearson and spearman correlations are good metrics for comparing the predicted and ground-truth values between two variables, they can be fooled easily with sparse data like gene expression data where non-zero values are sparse. We suggest computing other metrics like the Mean squared Error and R2 score (coefficient of determination) to support the claim.

The sCellST model performs competitively with other algorithms for spot level predictions: While the benchmark implementation itself is valid, the datasets used for the benchmark needs to be extensive to claim the competitive nature. Especially in these experiments, only the partial dataset of the kidney has been used. A total of 24 visium samples exist for visium from the cohort, however only 8 are used in a cross-validation setting. This is not explicitly stated in the paper but can be looked up in the slide ID metadata. Additionally, the samples of Prostate come from the same patient as suggested in the metadata of HEST. Further effort should be put into more detailed evaluations of external test sets to show the performance of the proposed model.

Line 588 - 593, the authors compare the coherence of the predictions from the DL network by comparing them to manual annotations on 2 slides, breast and ovarian cancer. While the SSL network won't have seen the gexp predictions, the details on how the sCellST model was trained for gexp prediction on these slides is missing. It's presumably trained on the given 2 slides and validated with experiments on the same slide. The experiments further for cell-type expressions have been performed very well with statistical tests however this needs to be done extensively on more samples of the same cancer type which exist in the HEST database.

Section 2.5 is well written.

Section 2.6 - While the qualitative evaluation of the finer cell types in ovarian cancer is shown and plausible, the experiment has again only been performed on a single sample A more detailed evaluation of the datasets with statistical tests is required.

Presentation

The paper is well written and easy to understand. However, in many places some details are missing or obscured. In some experiments (Figures 3d-e), the authors trained and evaluated their model on slides from the same patients, which is not explicitly stated in the paper but can be looked up in the slide ID metadata. And in 361-362 it appears that one-hot encoded vectors outperform their feature extractors, but this was brushed aside as a side note.

Structure

The authors first demonstrated their method on a simulated dataset, followed by experiments on Visium and Xenium datasets. As mentioned in the analysis above, the simulated expressions should be discarded and more effort should be put into a more detailed evaluation of the real datasets.

(Remarks on code availability)

I did not run the code, but I did take a look at github, which has enough documentation on the code and it seems well-written.

Reviewer #4

(Remarks to the Author)

(Remarks on code availability)

The code is hosted on GitHub, but could be made more user-friendly.

Reviewer #5

(Remarks to the Author)

(Remarks on code availability)

Version 2:

Reviewer comments:

Reviewer #1

(Remarks to the Author)

The authors' revision and response addressed my concerns.

(Remarks on code availability)

I did not run the code, but it would be helpful if the authors included example data so users can test the code on a small sample and on their own datasets.

Reviewer #2

(Remarks to the Author)

The authors have responded to the previous comments. The reviewer agrees the article has achieved the level of journal quality requirements. As a result, the article is recommended to be published.

(Remarks on code availability)

Reviewer #3

(Remarks to the Author)

The authors have satisfactorily answered all our questions including details and requested experiments that strengthen the credibility of their approach. With these clarifications, we find the work of sCellST novel and promising in predicting single cell gene expression from HE images. The manuscript is also clearly written and accessible. We particularly appreciate the inclusion of additional validation samples and a detailed list of those used in the study.

That said, the manuscript continues to present strong claims regarding accurate gene expression prediction, while still relying on certain simplified assumptions (e.g., performance being more consistent with FFPE samples or under conditions where CellViT performs well). While we recognize that such constraints stem from the limitations of existing methods rather than this work itself, we encourage the authors to highlight these assumptions and potential batch effects more explicitly. Emphasizing these aspects would improve transparency and also help position sCellST as a strong foundation for future research aimed at extending its applicability beyond idealized settings. For example, we suggest that the authors should add the detailed responses to R1.5, R1.8, and R3.5 in the discussion section which addresses three major concerns all reviewers have.

(Remarks on code availability)

The documentation of the code and user support seems to be well-prepared in the github page, though we did not run the code ourselves.

Reviewer #4

(Remarks to the Author)

(Remarks on code availability)

The code has enough documentation, and it seems well-written.

Reviewer #5

(Remarks to the Author)

I co-reviewed this manuscript with one of the reviewers who provided the listed reports. This is part of the Nature

Communications initiative to facilitate training in peer review and to provide appropriate recognition for Early Career Researchers who co-review manuscripts.

(Remarks on code availability)

The code is hosted on GitHub, but could be made more user-friendly by adding tutorials

Revision Plan – Point-to-point answer to the reviewers

First of all, we would like to thank all the reviewers for their careful reading and insightful feedback on our manuscript. We have carefully addressed all concerns raised. Please find our point-by-point answers below.

Reviewer #1 (Remarks to the Author):

The study presents an innovative computational pathology method (sCellST) with potential for predicting single-cell gene expression from H&E images. It contributes valuable insights into the link between morphology and transcriptomics, with applications in large-scale tissue studies.

We would like to thank the reviewer for their positive feedback highlighting the novelty and potential impact of our approach.

However, I recommend to address the following concerns to enhance this study.

R1.1 The introduction is well-motivated, but it repeats background information on spatial transcriptomics in multiple places. The discussion of spatial transcriptomics (ST) vs. image-based ST methods could be more concise.

We thank the reviewer for their positive feedback on the introduction in general. We have shortened the paragraph on ST techniques, while retaining essential context for the reader.

R1.2 The MIL framework is well explained, but the notations are somewhat dense and scattered across sections. A single table summarizing key mathematical terms (e.g., GE prediction functions, loss functions, MIL assumptions) would improve clarity. Self-supervised learning (SSL) details should be clearly separated from MIL in a distinct subsection for easier comprehension.

We appreciate the reviewer's positive remarks on our explanation of the MIL framework. We agree that the notations can appear dense. We had already a list of notations in the methods section ("Notations and abbreviations"), which we have now completed to be more comprehensive. We put details about self-supervised learning and multiple instance learning in separate sections of the methods. However, for the main text, we cannot separate into more subsections because of the journal formatting rules.

R1.3 Some figures lack clear p-values or confidence intervals, which should be added for robustness. Also, figure legends should clearly explain the key comparisons (e.g., benchmarks across models, cell-type predictions).

We thank the reviewer for their comment. Indeed, we did not provide p-values systematically. In some cases—such as Figures 2b and 2c—the differences are so clear that we believe p-values would not add meaningful information. In other cases—such as Figures 2d and 2e—there are only small differences between MIL and supervised learning, and between SSL embeddings and one-hot encodings, respectively. Regardless of statistical significance, these differences suggest that the MIL approach performs comparably to full supervision, and that the SSL embeddings approximate the (ideal) performance of one-hot encodings. Again, in these specific instances, we believe that including p-values would not substantially enhance interpretation. Also, p-values are not always best for interpretation since with large sample size, even a very small difference becomes statistically significant while conveying no biological meaning as discussed in (Fay et al, 2018). In answer to this concern, we performed statistical analysis for figures 4e, 4f and 7b using Mann-Whitney-Wilcoxon two-sided test with Bonferroni correction (****: $p \leq 1.00e-04$). Because of the large number of cells, all experiments were statistically significant, as shown below. We believe that in this case, the box-plots may convey a better assessment for the distributional overlaps than the p-value.

Fay, David S., and Ken Gerow. “A Biologist’s Guide to Statistical Thinking and Analysis.” In *WormBook: The Online Review of C. Elegans Biology* [Internet]. WormBook, 2018. <https://www.ncbi.nlm.nih.gov/books/NBK153593/>.

R1.4 The study primarily reports Pearson/Spearman correlations for performance evaluation. However, additional metrics such as Mean Absolute Error (MAE), Root Mean Square Error (RMSE), or AUROC could further validate the model.

We thank the reviewer for this helpful suggestion. We included a *Model Evaluation* section in the methods where we discussed the choice of metrics to evaluate the models. In addition to Pearson and Spearman correlations, we included Mean Absolute Error (MAE) and Root Mean Square Error (RMSE) as complementary evaluation metrics for a subpart of the benchmark highlighting the robustness of correlation based metrics.

R1.5 The study lacks an explicit analysis of inter-slide variability, how well does the model generalize across datasets with different staining protocols?

We thank the reviewer for highlighting the important issue of inter-slide variability and generalization across datasets with differing staining protocols. While we agree that analyzing domain shifts is an important direction (as we discussed page 22, line 979), our current framework is not built on a foundation model and therefore does not benefit from the broader generalization capabilities such models aim to provide—even though recent studies suggest that foundation models are still affected by staining variability (Kömen et al., 2024). Addressing variability introduced by different staining protocols would require further dedicated investigation, including domain adaptation strategies, which we consider an important avenue for future work. These approaches would presumably also require much more data than what is currently available. We have outlined this in the Discussion of our article (Line 939-944).

Kömen, Jonah, Hannah Marienwald, Jonas Dippel, and Julius Hense. “Do Histopathological Foundation Models Eliminate Batch Effects? A Comparative Study.” arXiv, November 8, 2024. <http://arxiv.org/abs/2411.05489>.

R1.6 It would be helpful to include an ablation study to assess the contribution of different components (e.g., how much does SSL improve performance over a simple MIL baseline?).

We thank the reviewer for this comment which we address in the revised version of the manuscript by comparing two encoders in sCellST: the current version (using SSL) and an alternative where the encoder is pre-trained on ImageNet. The results are shown in SupFig.2 and referenced in the main text (page 12, line 519). We do not consider an approach where the MIL is trained directly from cell images since it is computationally very expensive and the field now commonly uses pretrained encoder such as CTransPath (Wang et al, 2022) or UNI (Chen et al 2024) for patch level representation. As we are working with much smaller single-cell images, these foundation models are unfortunately not suitable. We therefore restricted the analysis to self-supervised learning and ImageNet-pretraining.

Chen, Richard J., Tong Ding, Ming Y. Lu, Drew F. K. Williamson, Guillaume Jaume, Andrew H. Song, Bowen Chen, et al. “Towards a General-Purpose Foundation Model for Computational Pathology.” *Nature Medicine* 30, no. 3 (March 2024): 850–62. <https://doi.org/10.1038/s41591-024-02857-3>.

Wang, Xiyue, Sen Yang, Jun Zhang, Minghui Wang, Jing Zhang, Wei Yang, Junzhou Huang, and Xiao Han. “Transformer-Based Unsupervised Contrastive Learning for Histopathological Image Classification.” *Medical Image Analysis* 81 (October 1, 2022): 102559. <https://doi.org/10.1016/j.media.2022.102559>.

R1.7 The manuscript mentions using Highly Variable Genes (HVGs) and Spatially Variable Genes (SVGs) but does not clearly justify the choice of cutoff thresholds. A sensitivity analysis on different gene selection criteria (e.g., top 1000 vs. top 500 genes) would strengthen the findings.

We thank the reviewer for their comment and fully agree that the choice was arbitrary. While some form of gene selection is necessary - since the network cannot be trained on genes that lack variability - there is no theoretical basis for determining the cut-off. However, as suggested by the reviewer, we propose to investigate how the prediction metrics evolve with respect to the number of selected genes. We show the results in SupFig.1 and in the main text (page 12, line 512). While there is a slight trend that the best

predicted genes correspond to lower rank in the list, it seems that rank and prediction accuracy are only weakly related.

R1.8 The study demonstrates that sCellST can predict known cell types, but it does not explore novel biological insights gained from these predictions. Could sCellST uncover previously unrecognized morphological-genetic associations in tumors?

We agree with the reviewer that this is an exciting perspective which we have also considered. However, we believe that pursuing this direction might be premature at this stage. The major focus of our work is to demonstrate that sCellST can assign single-cell gene expression to individual cells based on their morphology alone, as captured in H&E-stained tissues. By showing that known cell morphologies can be automatically retrieved and linked to transcriptomic profiles, we believe that sCellST already makes a compelling case.

While the discovery of novel morphologies or new morphology-transcriptomic relationships would indeed be intriguing, it poses significant challenges, particularly due to the lack of prior knowledge about these morphologies. Validating whether such findings reflect true biological signals would be complex and would likely require additional orthogonal data.

Reviewer #1 (Remarks on code availability):

The codes are prepared well for users.

Reviewer #2 (Remarks to the Author):

* What are the noteworthy results? *

Predicting gene expression directly from histology is an active research area, and the authors position sCellST against two broad classes of existing approaches. Within this landscape, sCellST offers a unique contribution by delivering single-cell resolution predictions via a MIL paradigm, indicating the novelty of this work.

We thank the reviewer for this positive assessment of the novelty and relevance of the presented approach.

* Will the work be of significance to the field and related fields? How does it compare to the established literature? If the work is not original, please provide relevant references. *

Although there are similar ideas being published previously (e.g., STAnD [2]), but the authors of this manuscript have conducted a comprehensive research focusing on establishing a pipeline with more advanced approaches for predicting gene expression using single cell H&E images.

* Does the work support the conclusions and claims, or is additional evidence needed? *

Yes, the work supports the conclusions and claims

* Are there any flaws in the data analysis, interpretation and conclusions? Do these prohibit publication or require revision? *

To the reviewer's point of view, the correlation between predicted gene expression and actual gene expression is a critical information for evaluating the performance of the proposed pipeline. The authors are suggested to provide this comparison.

We fully agree with the reviewer that correlation between predicted gene expression and actual gene expression is an important metric to report. For this reason, we report correlation in Figure 2 for our simulation study and in Figure 3.c and 3.e as the main metric for benchmarking methods on the kidney and prostate datasets (patch-level). At the single-cell level, this metric is difficult to obtain, as for Visium data, we do not have single-cell gene expression data. For this reason, we turned to a dataset for which we have Visium and Xenium data and provide correlations of gene expression between actual and predicted gene expression at the single cell level in Figure 6.d. We have added a table as extended data, referenced in the main text (page 18, line 783), showing gene expression correlations across all Xenium datasets evaluated. However, these values should be

interpreted with caution, as they are highly dependent on the training data and the specific validation slide used.

* Is the methodology sound? Does the work meet the expected standards in your field? *

Yes

* Is there enough detail provided in the methods for the work to be reproduced? *

Yes

Reviewer's comment:

The authors propose sCellST, a pipeline that predicts gene expression at single-cell resolution from H&E-stained tissue images by training on paired spatial transcriptomics (ST) data. The methodology is clearly structured into three main components: (1) Cell Detection and Segmentation, (2) Self-Supervised Feature Extraction, and (3) Multiple Instance Learning (MIL) gene expression prediction as shown in Fig.1 of the manuscript. Overall, the approach is logically sound and well-described, with each step justified by prior work or preliminary experiments.

The authors anticipate the main challenges (domain shift and segmentation/classification errors) and discuss them. The model's performance on single-cell data (Xenium) and across multiple tissues indicates its performance and can be adapted relatively easily (as the result has suggested the overfit is contaminated). With more training data and perhaps minor methodological tweaks (domain augmentation), sCellST could likely be deployed widely.

We thank the reviewer for this positive assessment of the article, regarding both the method design and the description.

Below are some points that the reviewer

R2.1 In the article, the authors stated that "CellVIT, predicts six distinct classes: neoplastic epithelial cells, connective/soft-tissue cells (including fibroblasts, muscle, and endothelial nuclei), inflammatory cells, non-neoplastic epithelial cells, dead cells, and unlabelled cells. In our analysis of the H&E slides, we observed that the majority of cells were classified into one of the first three categories (see SupTable.1). Therefore, we restricted our analysis to these three primary classes." I agree that cancer cells, connective cells and inflammatory cells are possibly the easiest 3 cell categories which is able to be classified at single cell level, while other cell types could be more challenging (although pathologists may be able to classify cell types based on bigger picture rather than using nuclei images).

The author may want to evaluate the impacts of using a different cell detection / classification approach? E.g., the mentioned hover-net.

The first part of the comment relates to the number of classes used in our analysis. As explained in the manuscript, we limited our focus to three classes—neoplastic epithelial cells, connective/soft-tissue cells, and inflammatory cells—because the number of nuclei classified into the other categories (dead cells, non-neoplastic epithelial cells, unlabeled cells) was very low (as shown in SupTable.1). To ensure meaningful statistical analysis, we retained only the classes with sufficient number of nuclei. However, we also demonstrate in Figure 7.c that sCellST is capable of identifying finer categories, such as plasma cells and endothelial cells, not provided by CellViT or HoverNet.

The second part of the comment relates to the impact of the cell detection and classification method. We fully acknowledge that detection or segmentation quality play a critical role and we have discussed this limitation explicitly (page 22, lines 972). Our choice to use CellViT was based on the benchmark results reported in Hörst et al. [21], as well as its availability within HEST [33]. Importantly, classification results are not used by our method except for the validation step. Also, the classes used in HoverNet are very similar to those in CellViT.

To ensure robustness of our results with respect to cell detection/segmentation, we replicated the experience from the section “*sCellST can identify finer cell types in ovarian cancer*” using HoverFast, a faster implementation of HoverNet. As can be seen in SupFig 7., the galleries of images allowed to highlight similar cell morphology patterns.

R2.2 Although the authors have discussed the prediction performance at bag and instance level (via the perspective of MIL) in Fig 6. However, biologists would be more interested on the correlation between the predicted gene “expression” v.s. the actual gene expression. E.g., computing the “pseudo spot” similar to [2] and the spearman correlation. This approach will be able to provide not only the gene list of high correlations (as shown in Fig. 7) but also provide the level of correlation in the ranked gene list.

We thank the reviewer for their comment. We have provided correlations between predicted gene expression and actual gene expression as suggested by the reviewer in Figures 2, 3 and 6. In addition and in answer to the reviewer’s request, we now provide SupData.1 of ranked gene lists indicating the correlations referenced in the main text (page 18, line 783), such that it becomes clear to the reader which gene expression is predictable with our model and to which extent. However, these values should be interpreted with caution, as they are highly dependent on the training data and the specific validation slide used.

R2. 3. Single-cell level gene expression prediction represents a great challenge. Various researchers have tackled this challenge, e.g., XFuse [1], aims to predict the whole slide gene expression at pixel level. STAnD [2] predicting single-cell gene expression based on H&E using MIL (which is similar to the idea proposed in this manuscript).

We understand this comment as a request to add missing references, which we are happy to provide (XFuse was already cited). We have added the references to the revised version of the article (page 4, line 159).

R2. 4. Some references are missing, e.g., Transfer learning, ImageNet, Self-supervised learning and, ResNet-50 (page 23, line 1055).

We have added the missing references to the revised version of the article (page 25, line 1133).

R2. 5. There are some duplicated acronyms, e.g., SSL in page 5, line 228 and SSL in page 23 line 1055.

We thank the reviewer for pointing out the duplicated acronym usage. The manuscript has been revised accordingly.

R2. 6. The authors are suggested to revise the manuscript more carefully. Although the reviewer didn't have spotted major error in the manuscript, but there are some errors, misspelling, etc., have to be addressed., including but not limited to line 1273, page 28, CellViT (is it CellViT with the lower case "i")

We thank the reviewer for pointing out the duplicated acronym usage. The manuscript has been revised accordingly.

REFERENCES

[1] Bergensträhle, L., He, B., Bergensträhle, J. et al. Super-resolved spatial transcriptomics by deep data fusion. Nat Biotechnol 40, 476–479 (2022). <https://doi.org/10.1038/s41587-021-01075-3>

[2] Huang C. H., Park Y.; Pang, J., Bienkowska, J. R., Single-cell gene expression prediction using H&E images based on spatial transcriptomics, Proceedings Volume 12471, Medical Imaging 2023: Digital and Computational Pathology; 1247105 (2023) <https://doi.org/10.1117/12.2654294>

Reviewer #3 (Remarks to the Author):

Summary

The paper proposes a new method for predicting gene expressions from histology images. The method is based on multiple instance learning on a bag of features derived from single cell images. The images come from a cell segmentation model and the features are obtained from a self-supervised feature extractor. Despite being trained on spot expressions, the model appears to be able to predict single-cell expressions, which is novel.

We thank the reviewer for this positive assessment of the novelty of the presented approach.

Although the method has been evaluated on several tasks and both simulated and real datasets, the results presented are not sufficient to validate the strong claims made throughout the paper, see below for detailed comments.

We thank the reviewer for the critical reading and we are happy to address all the points raised below.

Analysis

The prediction of single cell expressions is an interesting and valuable topic. Since there is no ground-truth available for the single cell expressions paired with histology images in the Visium samples, the paper suggests using the CellViT segmentation labels and their corresponding annotated scRNA-seq expressions together to get a broad perspective of the expressed genes for each cell in histology. While this is a well thought approach several questions arise in various experiments which are crucial to support the credibility of these claims. We have listed them in the following points:

We thank the reviewer for this analysis. We would like to add that simulation is one among several validation strategies employed throughout the paper.

R3.1 Section 2.2 - The paper states it can predict the single cell level gene-expression in simulated data. This is shown with an example from an ovarian cancer visium slide and an annotated single cell reference taken from CellXGene. However the details on the biological reasoning of how the image and single cell expression clusters are matched are missing. For example, in the 312-314 centroid scenario, it's unclear how the clusters were matched (i.e what is the corresponding scRNA-seq cluster?). Similarly 315-317 what is the corresponding morphological cluster and how is it decided to match the image and scRNA

clusters? Further details need to be added to line 1228-1231 to support this experiment. Additionally, the simulation experiment has been performed only on a single visium slide of ovarian cancer which limits the heterogeneity observed across visium samples and spatial-omics in general. The experiments need to be done on multiple samples to support the claim.

We fully agree with the reviewer that the use of simulations for the validation of algorithms is always limited by the way in which the simulation is conducted.

On the other hand, simulations can provide an important validation tool, as they allow to benchmark certain design choices in a controlled environment. And this is how we used simulations here. We regret if not all the aspects of the simulation were entirely clear, and we added a few clarifications.

In brief, the association of image and scRNAseq clusters was arbitrary and did not reflect a biological reality. Our simulation aimed at testing algorithmic performance in a case where there was a clear (yet arbitrary) cluster correspondence between morphological and transcriptomic data. For this, we generated simulated data such that two cells clustering together in morphological space also belonged to the same scRNAseq cluster. While this is a strong hypothesis (and ultimately not realistic), this scenario allowed us to test whether the MIL approach we developed could – in such a case – retrieve single cell gene expressions. Of note, our simulation scenario was at the same time optimistic (as there was a clear correspondence between morphology and molecular profiles) and pessimistic (as there was no intra-cluster relationship between transcriptomic and morphology profiles, by construction).

As we always have – by construction – transcriptomic and morphological clusters that are arbitrarily matched, we do not believe that extending the simulations to more slides would provide more insights. We added some sentences to better explain the simulations, both in the main text (page 7, lines 297, 315, 324) and the simulation subsection of the method section.

R3.2 Line 326 - 355 shows high mean correlation between the ground-truth spot and predicted aggregate of cell expressions for the top 1000 highly variable genes. This experiment is done to show that the MIL approach can recover GE vectors from cell images. It's important to note, it is quite ambiguous in this section as well as in the methods section whether the HVGs are computed on the Visium samples or single cell atlas. This is crucial since it's not necessary that the spot and cellular level HVGs are identical.

In the simulation section, we compute highly variable genes (HVGs) from the single-cell RNA-seq dataset (as stated in the methods section) to ensure a consistent gene set across all experiments. This approach allows us to standardize gene selection despite variations in the training sets introduced by different simulation settings. In the other experiments of the manuscript, HVGs are instead computed directly from the Visium dataset. We have provided some additional explanation in the revised version of the manuscript in the *ST preprocessing subsection, Methods section*.

R3.3 While Pearson and Spearman correlations are good metrics for comparing the predicted and ground-truth values between two variables, they can be fooled easily with sparse data like gene expression data where non-zero values are sparse. We suggest computing other metrics like the Mean squared Error and R2 score (coefficient of determination) to support the claim.

We thank the reviewer for this insightful comment. We included a “Model Evaluation” section in the methods where we discussed the choice of metrics to evaluate the models. In addition to Pearson and Spearman correlations, we included Mean Absolute Error (MAE) and Root Mean Square Error (RMSE) as complementary evaluation metrics for a subpart of the benchmark highlighting the robustness of correlation-based metrics. We included a comparison of the behavior of these metrics on a subset of the benchmark dataset in SupFig.9 and it can be seen that all metrics show the same ordering.

We chose not to include the R2 score among our evaluation metrics, as it implicitly assumes a linear relationship with a slope of 1 in order to yield high values. This assumption does not hold in our setting due to batch effects across slides, which can shift the scale of gene expression measurements without affecting their relative patterns. As discussed in the Model Evaluation section, correlation-based metrics such as PCC and SCC are more appropriate in this context. PCC is also the reference metric used in the HEST database to evaluate Foundation Models on the prediction of gene expression from tile images.

R3.4 The sCellST model performs competitively with other algorithms for spot level predictions: While the benchmark implementation itself is valid, the datasets used for the benchmark needs to be extensive to claim the competitive nature. Especially in these experiments, only the partial dataset of the kidney has been used. A total of 24 visium samples exist for visium from the cohort, however only 8 are used in a cross-validation setting. This is not explicitly stated in the paper but can be looked up in the slide ID metadata. Additionally, the samples of Prostate come from the same patient as suggested in the metadata of HEST. Further effort should be put into more detailed evaluations of external test sets to show the performance of the proposed model.

We thank the reviewer for this detailed comment, and we thank the reviewer for agreeing with the benchmark implementation. We added more information in the *data availability section*. For all Visium samples, we intentionally excluded frozen sections, as the quality of nuclear morphology is compromised, and we also discarded samples with staining artifacts that interfered with accurate cell segmentation—an essential component of our pipeline. Here is an example of the difference between H&E from Frozen and FFPE samples.

Crop of FFPE slide

Crop of Frozen slide

Finally, we also excluded some FFPE slides due to poor quality. Indeed, the H&E slides acquired along with ST data tends to be very heterogeneous in quality, and for our approach, we require high quality H&E slides. As shown in the following plots, the four excluded slides exhibit poor staining quality, making reliable nucleus segmentation unfeasible with current state-of-the-art methods. In such conditions, any method based on nuclear segmentation is at high risk of providing unreliable results. This limitation of our method is explicitly mentioned in the discussion (page 22, lines 970-978).

Number of detected cells in spots for FFPE Kidney dataset

Number of detected cells in spots for FFPE Kidney dataset

Regarding the prostate dataset, while the samples originate from the same patient, we would like to stress that this is consistent across all compared methods and does not confer any advantage to our model.

Finally, we would like to emphasize that the primary goal of our method is to predict gene expression at the single-cell level. However, few existing methods address this specific problem, and those with broader visibility typically focus on spot-level prediction. In this context, and for the purpose of benchmarking, we chose to position our method relative to spot-level approaches—because this is the task for which several established methods exist and for which ground truth data is available. Our aim was to assess whether our method achieves comparable overall performance for this related but not identical task. However, we do not claim that our method outperforms existing tools for this specific task. Indeed, if the objective is to predict spot-level gene expression, it would likely be more appropriate to use a model specifically trained for that purpose. Since this benchmarking does not directly align with the primary objective of our method, we believe that broader validation with additional data would not substantially strengthen the paper.

R3.5 Line 588 - 593, the authors compare the coherence of the predictions from the DL network by comparing them to manual annotations on 2 slides, breast and ovarian cancer. While the SSL network won't have seen the gexp predictions, the details on how the sCellST model was trained for gexp prediction on these slides is missing. It's presumably trained on the given 2 slides and validated with experiments on the same slide. The experiments further for cell-type expressions have been performed very well with statistical tests however this needs to be done extensively on more samples of the same cancer type which exist in the HEST database.

We thank the reviewer for this comment. We have now clarified this aspect in the main text (page 13, line 574).

We would like to stress that this comparison serves as a sanity check, helping confirm that our weakly supervised framework can produce coherent predictions even without explicit cell-level supervision.

That said, we acknowledge the limitations of this setup, as CellViT only outputs very broad cell types. For this reason, we preferred to spend more time on comparing our predictions to high-resolution spatial transcriptomics data (e.g. Xenium, see Figure 6), which we believe provide stronger evidence of the model's abilities.

R3.6 Section 2.5 is well written.

R3.7 Section 2.6 - While the qualitative evaluation of the finer cell types in ovarian cancer is shown and plausible, the experiment has again only been performed on a single sample. A more detailed evaluation of the datasets with statistical tests is required.

We thank the reviewer for this suggestion. First, we understand the request for statistical testing. The problem here is that the number of cells is very high, and in this setting, a statistical test tends to always indicate significance. Indeed, under a setting with a large sample size, even a very small difference becomes statistically significant while conveying no biological meaning as discussed in (Fay et al, 2018). In answer to this concern, we performed statistical analysis for figure 7b using Mann-Whitney-Wilcoxon two-sided test with Bonferroni correction (****: $p \leq 1.00e-04$). Because of the large number of cells, all experiments were statistically significant, as shown below. We believe that in this case, the box-plots may convey a better assessment for the distributional overlaps than the p-value.

Fay, David S., and Ken Gerow. "A Biologist's Guide to Statistical Thinking and Analysis." In *WormBook: The Online Review of C. Elegans Biology* [Internet]. WormBook, 2018. <https://www.ncbi.nlm.nih.gov/books/NBK153593/>.

Second, the reviewer is concerned about the results originating only from a single slide. To constructively address this comment, we applied the same algorithm to a breast cancer slide. The results are reported in SupFig.6 and discussed in the main text (page 20, line 875).

Presentation

R3.8 The paper is well written and easy to understand. However, in many places some details are missing or obscured. In some experiments (Figures 3d-e), the authors trained and evaluated their model on slides from the same patients, which is not explicitly stated in the paper but can be looked up in the slide ID metadata. And in 361-362 it appears that one-hot encoded vectors outperform their feature extractors, but this was brushed aside as a side note.

We thank the reviewer for the overall positive assessment and these thoughtful observations.

Regarding the point that the model was trained and evaluated on slides from the same patients, we note that this setup is applied consistently across all compared models and does not provide a particular advantage to our approach.

As for the observation about the one-hot encoded vectors (page 10, line 415), we appreciate the opportunity to clarify this point. As mentioned in page 10, line 417, these vectors were used only in the simulation setting and represent an idealized, noise-free embedding that perfectly reflects the underlying image clusters. This design choice intentionally defines an upper-bound scenario (only in the simulation setting), and the similar performance of the SSL features compared to the one-hot-encoded vectors highlights the ability of our model to capture the meaningful information even when hidden in a large input vector. We added more details in the simulation subsection methods and hope that it is clearer in the revised version of our manuscript.

Structure

R3.9 The authors first demonstrated their method on a simulated dataset, followed by experiments on Visium and Xenium datasets. As mentioned in the analysis above, the simulated expressions should be discarded and more effort should be put into a more detailed evaluation of the real datasets.

We thank the reviewer for the suggestion. While we understand the concern about the use of simulated data, we would like to clarify that the simulation experiment was not intended to substitute for real data evaluation, but rather to offer an interpretable, controlled setting where ground-truth expression at the cell level is known. This enables us to evaluate the model's behavior in a setting where each component can be explicitly validated—something not feasible with real-world datasets due to the lack of ground truth at the single-cell level. In particular, we use the simulation framework to investigate whether the

MIL strategy is in principle applicable to this problem under the favorable hypothesis that there is a match between morphological and transcriptomic clusters.

Regarding evaluation on real datasets, we would like to highlight that our study includes a comprehensive assessment across multiple tissue types and platforms. Specifically, we applied our approach on a total of 15 Visium slides spanning 4 datasets, and now 9 Xenium slides from 4 different datasets. This is in line with or exceeds the scale of many recently published methods, as summarized in the table below. We distinguish older spatial transcriptomic slides (prior to Visium) and/or frozen slides, which are not considered in this study for the reasons mentioned earlier.

We included more Xenium slides to reach 9 which is more than all the other published articles to our knowledge.

Dataset numbers: a comparison in different studies

Reference	# Datasets / # Visium FFPE slides (# Number of slides per dataset)	# Datasets / # Visium Frozen slides (# Number of slides per dataset) Unusable	# Datasets / # Spatial Transcriptomic s (# Number of slides per dataset) Unusable	# Datasets (# Xenium slides)(# Number of slides per dataset)
Ours	4 / 15 (8 / 5 / 1 / 1)	0 / 0	0 / 0	9 / 4 (3 / 2 / 2 / 2)
XFuse (Nat. Methods)	0 / 0	3 / 7 (2 / 4 / 1)	12 / 1 (12)	0
IStar (Nat. Biotechnolo gy)	3 / 3 (1 / 1 / 1)	6 / 6 (1 / 1 / 1 / 1 / 1 / 1)	1 / 2 (2)	2 / 3 (2 / 1)
THltoGene (Briefing in Bioinformati cs)	0 / 0	0 / 0	2 / 44 (32 / 12)	0
McIStExp (Briefing in Bioinformati cs)	1 / 1 (1)	1 / 8 (8)	2 / 44 (32 / 12)	0

Reviewer #3 (Remarks on code availability):

I did not run the code, but I did take a look at github, which has enough documentation on the code and it seems well-written.

Reviewer #4 (Remarks to the Author):

Reviewer #4 (Remarks on code availability):

The code is hosted on GitHub, but could be made more user-friendly.

Reviewer #5 (Remarks to the Author):

Revision Plan – Point-to-point answer to the reviewers - 2

Reviewer #1 (Remarks to the Author):

The authors' revision and response addressed my concerns.

Reviewer #1 (Remarks on code availability):

I did not run the code, but it would be helpful if the authors included example data so users can test the code on a small sample and on their own datasets.

Reviewer #2 (Remarks to the Author):

The authors have responded to the previous comments. The reviewer agrees the article has achieved the level of journal quality requirements. As a result, the article is recommended to be published.

Reviewer #3 (Remarks to the Author):

The authors have satisfactorily answered all our questions including details and requested experiments that strengthen the credibility of their approach. With these clarifications, we find the work of sCellST novel and promising in predicting single cell gene expression from HE images. The manuscript is also clearly written and accessible. We particularly appreciate the inclusion of additional validation samples and a detailed list of those used in the study.

That said, the manuscript continues to present strong claims regarding accurate gene expression prediction, while still relying on certain simplified assumptions (e.g., performance being more consistent with FFPE samples or under conditions where CellViT performs well). While we recognize that such constraints stem from the limitations of existing methods rather than this work itself, we encourage the authors to highlight these assumptions and potential batch effects more explicitly. Emphasizing these aspects

would improve transparency and also help position sCellST as a strong foundation for future research aimed at extending its applicability beyond idealized settings. For example, we suggest that the authors should add the detailed responses to R1.5, R1.8, and R3.5 in the discussion section which addresses three major concerns all reviewers have.

We thank the reviewer for this comment. We have now clarified this aspect in the main text, in the discussion section.

Reviewer #3 (Remarks on code availability):

The documentation of the code and user support seems to be well-prepared in the github page, though we did not run the code ourselves.

Reviewer #4 (Remarks to the Author):

Reviewer #4 (Remarks on code availability):

The code has enough documentation, and it seems well-written.

Reviewer #5 (Remarks to the Author):

Reviewer #5 (Remarks on code availability):

The code is hosted on GitHub, but could be made more user-friendly by adding tutorials